# Reformulation of an extant ATPase active site to mimic ancestral GTPase activity reveals a nucleotide base requirement for function

Taylor B Updegrove[1†], Jailynn Harke[1†], Vivek Anantharaman[2], Jin Yang[3], Nikhil Gopalan[1], Di Wu[4], Grzegorz Piszczek[4], David M Stevenson[3], Daniel Amador-Noguez[3], Jue D Wang[3], L Aravind[2], Kumaran S Ramamurthi[1*]

[1]Laboratory of Molecular Biology, National Cancer Institute, National Institutes of Health, Bethesda, United States; [2]National Center for Biotechnology Information, National Library of Medicine, National Institutes of Health, Bethesda, United States; [3]Department of Bacteriology, University of Wisconsin, Madison, United States; [4]Biophysics Core Facility, National Heart, Lung and Blood Institute, National Institutes of Health, Bethesda, United States

**Abstract** Hydrolysis of nucleoside triphosphates releases similar amounts of energy. However, ATP hydrolysis is typically used for energy-intensive reactions, whereas GTP hydrolysis typically functions as a switch. SpoIVA is a bacterial cytoskeletal protein that hydrolyzes ATP to polymerize irreversibly during *Bacillus subtilis* sporulation. SpoIVA evolved from a TRAFAC class of P-loop GTPases, but the evolutionary pressure that drove this change in nucleotide specificity is unclear. We therefore reengineered the nucleotide-binding pocket of SpoIVA to mimic its ancestral GTPase activity. SpoIVA[GTPase] functioned properly as a GTPase but failed to polymerize because it did not form an NDP-bound intermediate that we report is required for polymerization. Further, incubation of SpoIVA[GTPase] with limiting ATP did not promote efficient polymerization. This approach revealed that the nucleotide base, in addition to the energy released from hydrolysis, can be critical in specific biological functions. We also present data suggesting that increased levels of ATP relative to GTP at the end of sporulation was the evolutionary pressure that drove the change in nucleotide preference in SpoIVA.

*For correspondence:
ramamurthiks@mail.nih.gov

†These authors contributed equally to this work

Competing interests: The authors declare that no competing interests exist.

## Introduction

Nucleotides have various functions in the cell, as coenzymes, signaling messengers, and the building blocks of genetic material. Nucleoside triphosphates (NTPs) store energy in the form of their phosphate bonds. The free energy of the hydrolysis reaction involving the bond between the β- and the γ-phosphates is approximately 30 kJ/mol (*Berg et al., 2007*) and is used to drive a variety of energy-consuming biochemical reactions. Despite this similarity between different NTPs, enzymes usually display strong preferences toward a specific NTP. For example, adenosine triphosphate (ATP) in general is the principal source of energy in a cell and is used by motor proteins to perform work, whereas hydrolysis of guanosine triphosphate (GTP) typically functions as a timer or switch, such as in proteins involved in signal transduction (*Alberts, 2002*). One explanation for this dichotomy is that the relative intracellular abundance of ATP in a cell drove the evolution of motors to use it as an energy source (*Bennett et al., 2009*; *Rudoni et al., 2001*; *Traut, 1994*). Consistent with that notion, the eukaryotic ATPase motor proteins myosin and kinesin are evolutionarily members of the TRAFAC class of GTPases, having emerged from an ancestral GTPase, but have switched their nucleotide

**eLife digest** Living organisms need energy to stay alive; in cells, this energy is supplied in the form of a small molecule called adenosine triphosphate, or ATP, a nucleotide that stores energy in the bonds between its three phosphate groups. ATP is present in all living cells and is often referred to as the energy currency of the cell, because it can be easily stored and transported to where it is needed.

However, it is unknown why cells rely so heavily on ATP when a highly similar nucleotide called guanosine triphosphate, or GTP, could also act as an energy currency. There are several examples of proteins that originally used GTP and have since evolved to use ATP, but it is not clear why this switch occurred. One suggestion is that ATP is the more readily available nucleotide in the cell.

To test this hypothesis, Updegrove, Harke et al. studied a protein that helps bacteria transition into spores, which are hardier and can survive in extreme environments until conditions become favorable for bacteria to grow again. In modern bacteria, this protein uses ATP to provide energy, but it evolved from an ancestral protein that used GTP instead.

First, Updegrove, Harke et al. engineered the protein so that it became more similar to the ancestral protein and used GTP instead of ATP. When this was done, the protein gained the ability to break down GTP and release energy from it, but it no longer performed its enzymatic function. This suggests that both the energy released and the source of that energy are important for a protein's activity. Further analysis showed that the modern version of the protein has evolved to briefly hold on to ATP after releasing its energy, which did not happen with GTP in the modified protein.

Updegrove, Harke et al. also discovered that the levels of GTP in a bacterial cell fall as it transforms into a spore, while ATP levels remain relatively high. This suggests that ATP may indeed have become the source of energy of choice because it was more available.

These findings provide insights into how ATP became the energy currency in cells, and suggest that how ATP is bound by proteins can impact a protein's activity. Additionally, these experiments could help inform the development of drugs targeting proteins that bind nucleotides: it may be essential to consider the entirety of the binding event, and not just the release of energy.

specificity to utilizing the more abundant nucleotide ATP to perform their energy-intensive functions (*Leipe et al., 2002*).

In this report, we examine an unusual bacterial ATPase named SpoIVA (*Ramamurthi and Losick, 2008*; *Roels et al., 1992*) that is also from the TRAFAC class of GTPases and is exclusively found in sporulating members of the Firmicutes phylum (*Castaing et al., 2013*). We had previously shown that within the TRAFAC group, SpoIVA is closest to the Era GTPases (*Castaing et al., 2013*) which are switches involved in the maturation of 16S rRNA and assembly of the 30S ribosomal subunit and universally conserved across bacteria (*Ji, 2016*). Given the universal conservation of Era among bacteria and the narrow conservation of SpoIVA, we proposed a model in which SpoIVA emerged via a duplication event followed by rapid divergence from Era (*Castaing et al., 2013*). This divergence from Era included not only multiple residue substitutions, but also the addition of two C-terminal domains in SpoIVA that are not present in Era (*Figure 1B,C*). Further, the most parsimonious explanation of the phyletic patterns of SpoIVA and Era is that they are not formally sister groups where both diverged from a common ancestor; rather, SpoIVA ATPase, which emerged specifically in sporulating Firmicutes, has the Era GTPase itself as its ancestor. Unlike myosin and kinesin, SpoIVA is not a motor protein. Instead SpoIVA is a cytoskeletal protein that assembles into a static polymer in an ATP hydrolysis-dependent manner (*Ramamurthi and Losick, 2008*). In the absence of an obvious motor function, which would necessitate high ATP utilization, the selective pressure that drove the evolution of nucleotide preference in SpoIVA has been unclear.

SpoIVA is essential for bacterial endospore formation (*Galperin et al., 2012*; *Roels et al., 1992*). When *Bacillus subtilis* faces starvation, it metamorphoses into a structurally and chemically robust dormant cell type termed an endospore (hereafter a 'spore') that protects the cell's genetic material from environmental insults (*Higgins and Dworkin, 2012*; *Setlow, 2006*; *Stragier and Losick, 1996*; *Tan and Ramamurthi, 2014*). Spores are encased in a proteinaceous shell, termed the spore 'coat',

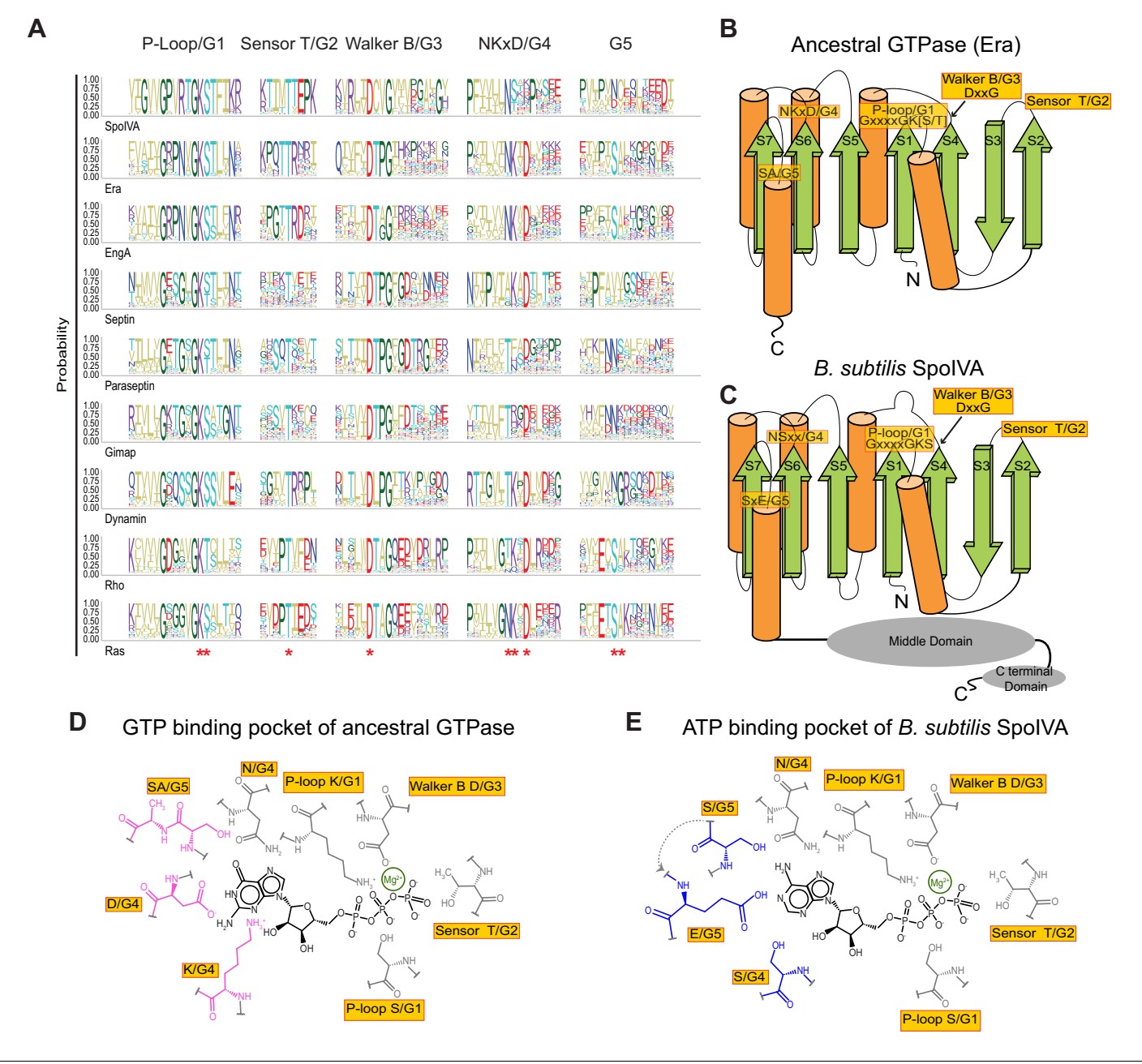

**Figure 1.** Predicted residues in SpoIVA evolved from an ancestral GTPase to bind ATP. (**A**) Sequence logo displaying conservation of amino acid residues in different members of the TRAFAC class of GTPases. Letters represent amino acid abbreviations; height of each letter represents the probability of conservation among orthologs of the indicated protein. Red asterisk below the sequence logo indicates absolute conservation of the amino acid at that position. (**B and C**) Topological representation of (**B**) ancestral TRAFAC GTPase or (**C**) SpoIVA. Motifs in the active site are indicated in yellow; numbering (G1–G5) corresponds to an idealized GTPase (*Bourne et al., 1991*). N: amino terminus; C: carboxy terminus. β-strands are depicted as green arrows; α-helices are depicted as orange cylinders. Middle and C-terminal domains of SpoIVA are depicted as gray ovals. (**D and E**) Depiction of the nucleotide-binding pocket of (**D**) ancestral TRAFAC GTPase bound to GTP or (**E**) SpoIVA bound to ATP. Residues in the ancestral GTPase that contact the guanine base of GTP are depicted in pink; predicted residues in SpoIVA that may bind the adenine base of ATP are depicted in blue.

a complex structure that is composed of ~80 proteins (*Driks and Eichenberger, 2016*; *Henriques and Moran, 2007*; *McKenney and Eichenberger, 2012*). Assembly of the coat begins with the construction of a basement layer, of which the major structural protein is SpoIVA (*McKenney et al., 2010*; *Peluso et al., 2019*; *Price and Losick, 1999*; *Ramamurthi et al., 2006*; *Ramamurthi and Losick, 2008*; *Roels et al., 1992*). Unlike dynamic cytoskeletal proteins like actin and tubulin, where nucleotide binding drives polymerization and nucleotide hydrolysis is linked to polymer disassembly (*Pollard and Goldman, 2018*), SpoIVA polymerization requires ATP hydrolysis to form a nucleotide-free static polymer (*Castaing et al., 2013*; *Wu et al., 2015*).

The amino terminal half of the SpoIVA ATPase is the nucleotide-binding domain, which belongs to the TRAFAC class of P-loop GTPases (*Figure 1A–C*; *Castaing et al., 2014*; *Castaing et al., 2013*; *Leipe et al., 2002*). This domain harbors a Walker A motif that binds the γ-phosphoryl group of the bound ATP (*Walker et al., 1982*), a Walker B motif that coordinates a $Mg^{2+}$ ion required for ATP hydrolysis and, like TRAFAC GTPases, a sensor Thr ('sensor T') that detects the γ-phosphoryl of the bound nucleotide to trigger ATP hydrolysis (*Leipe et al., 2002*). The classic TRAFAC GTPases which hydrolyze GTP contain a fourth motif (the so-called G4 motif), typically consisting of Asn and Lys separated by one residue before an Asp (NKxD) that confers guanine-binding specificity (*Leipe et al., 2002*). Crystal structures revealed direct interactions between the side chain of the Asp and the base of a bound GTP through two H-bonds, and a coordination of a water molecule to the α-phosphate group of the bound nucleotide by the Lys (*Knihtila et al., 2015*). Additionally, the extended aliphatic side chain of the said Lys forms a hydrophobic wall to hold the guanine of the nucleotide in the active site. Substitution of the Asp in this motif with Asn abolished the guanine-binding specificity of Ras, also a GTPase of the TRAFAC clade, and switched its specificity to xanthine (*Kang et al., 1994*; *Weijland et al., 1994*; *Zhong et al., 1995*). Additionally, in TRAFAC GTPases the loop spatially adjacent to the NKxD motif (the so-called G5 motif) is proximal to the guanine and might contribute to some extent to their guanine specificity.

To understand the functional requirement for the evolution of SpoIVA from a GTPase to an ATPase, we sought to reformulate the active site of SpoIVA to mimic its ancestral GTPase activity by restoring the NKxD motif, which is altered in SpoIVA, and by altering an SxE sequence in the loop associated with the G5 motif. We found that partial restoration of the NKxD motif and alteration of the SxE sequence resulted in a SpoIVA variant that hydrolyzed GTP in vitro slightly preferentially over ATP with reduced overall catalytic efficiency, similar to the in vitro activity of the Era GTPase from which SpoIVA was likely derived in the ancestral sporulating Firmicute (*Castaing et al., 2013*). In parallel, disrupting the NKxD motif in Era and introducing the SxE sequence into the G5 motif to mimic SpoIVA resulted in an Era variant that preferentially hydrolyzed ATP in vitro. The altered SpoIVA was able to harness the energy released by GTP or ATP hydrolysis to drive a necessary conformational change in the protein but failed to ultimately polymerize specifically in the presence of GTP. We show that this was due to the inability of SpoIVA to form the equivalent of an ADP-dependent multimer in the presence of GTP, which we show is a necessary intermediate en route to functional SpoIVA polymerization. Additionally, we provide evidence that the extant SpoIVA polymerizes more efficiently than the altered variant in the presence of lower ATP concentration, similar to what is present during the end of sporulation. We propose that a pronounced reduction in intracellular GTP concentration relative to ATP during the late stages of sporulation could have driven the evolution of SpoIVA to utilize ATP instead of GTP to drive polymerization, a critical step in the morphogenesis of the spore cell surface.

## Results

### Amino acid substitutions in the nucleotide-binding pocket could be responsible for the evolution of ATP-binding specificity in SpoIVA

Our earlier sequence-profile analysis along with site directed mutagenesis had shown SpoIVA to be a member of the TRAFAC class of GTPases with Era as its closest GTPase relative (*Castaing et al., 2013*). Since SpoIVA is strictly restricted to the clade of Firmicutes that form endospores we reasoned that SpoIVA was a relatively late innovation that evolved via gene duplication and rapid divergence from Era. Given that Era is a functional GTPase and SpoIVA shows a clear preference for ATP over GTP (*Ramamurthi and Losick, 2008*) we sought to identify the potential changes in the active

site that might have led to this shift in specificity. For relatively closely related proteins that diverged from a common ancestor, reconstruction of an ancestral state may be achieved based on phylogeny and using statistical methods across the entire length of the proteins (*Hochberg and Thornton, 2017*). However, the phyletic patterns strongly indicate that the ultimate ancestor of SpoIVA was Era itself. Moreover, SpoIVA and Era have diverged too far from each other to successfully employ such an approach. Beyond myriad amino acid substitutions and insertions, SpoIVA has even acquired two C-terminal domain fusions that are not present in Era (*Figure 1C*; *Castaing et al., 2014*), which is reminiscent of other examples where the acquisition of large appendages to ancestral proteins have generated novel functions beyond the ancestral function (*Escudero et al., 2020*; *Farr et al., 2017*). Thus, the number of variables involved when all residues are considered would result in too vast of a parameter space to analyze using a common ancestor reconstruction method. We therefore focused on the highly conserved N-terminal TRAFAC NTPase domain of SpoIVA and computed a sequence logo for SpoIVA, Era and various other families of the TRAFAC class of GTPases, especially those of the GIMAP-Septin-Dynamin clade which show a comparable tendency for forming oligomers or polymers (*Figure 1A*). Next, the amino acid conservation pattern of SpoIVA was superimposed on a topology diagram of the ancestral core TRAFAC GTPase domain (*Figure 1B*). This allowed us to identify those conserved active site positions which were retained in the ancestral state in SpoIVA and those that were altered with respect to *bona fide* GTPases (*Figure 1C*).

The first three motifs (G1–3) respectively correspond to: the Walker A motif which binds the triphosphate of the NTP substrate; the sensor T which discriminates the GTP-bound state from the GDP-bound state; and the Walker B motif which chelates the catalytic $Mg^{2+}$ and senses the bound triphosphate along with the sensor T (G2) motif (*Figure 1A,B*). In SpoIVA, these three motifs are retained in the ancestral state indicating that SpoIVA binds and senses the triphosphate moiety of the NTP similar to the ancestral GTPases (*Castaing et al., 2013*; *Figure 1B–E*).

In contrast, notable changes are seen in the G4 and G5 motifs of SpoIVA. Of these, G4 is comprised of the final residue of strand 6 of the core GTPase domain and a characteristic single-turn helix that follows it (*Figure 1B,C*). Among the GTPases closely related to SpoIVA, the G4 motif is of the form NKxD (where 'x' is any amino acid), for example, in Era and Eng (*Figure 1A*). The first position of this motif is typically either N or T in most families of the entire GTPase superfamily. Thus, SpoIVA retains the ancestral state in this position. This residue forms the 'lower wall' of the base-binding pocket of the active site and by itself does not appear to discriminate between the purines (*Figure 1D,E*). The next position is a K in most families of *bona fide* GTPases (*Figure 1A*) and the extended sidechain of this lysine forms the 'lateral wall' of the base-binding pocket (*Figure 1D*). Strikingly, this K is consistently substituted by an alcoholic (S/T) residue in the SpoIVA family. The next conserved position in G4 is the D, which is the most important determinant of guanine specificity (*Figure 1D*). In SpoIVA it is again mostly substituted by either of several polar residues, such as K, H, N, or R. Notably, unlike in *bona fide* GTPases this position is poorly constrained in SpoIVA, suggesting a relaxation of selection, which might have allowed the emergence of ATP selectivity.

The G5 motif follows immediately after strand-7 of the core GTPase domain and typically displays the motif SAx in classical TRAFAC GTPases. This region forms the wall of the base-binding pocket opposite to that formed by the conserved lysine in G4 (*Figure 1D*). In the case of SpoIVA the position corresponding to the conserved S in the G5 motif is less constrained and is usually either N, D, or S. The next residue is usually a cysteine in SpoIVA. Hence, this residue is likely to be similar to A with respect to its hydrophobicity and is not situated close to the distinguishing atoms of the purine base of the bound nucleotide. Thus, it is unlikely to have a major effect on base selectivity. Two residues downstream of S, there is a position that contributes to the wall of the base-binding pocket. This position is not particularly conserved in TRAFAC GTPases as a whole but in SpoIVA is either acidic (D/E) or Q/N in 41% of the orthologs. Together, these observations suggested that the changes in the G4 (NKxD) and G5 (SxE) regions relative to the bona fide GTPases may have contributed to the emergence of ATP-specificity in SpoIVA (*Figure 1E*).

## The altered NKxD motif in G4 and SxE sequence in G5 mediate nucleotide specificity of SpoIVA

To determine the relative contributions of the altered NKxD and SxE motifs in G4 and G5, respectively, on nucleotide hydrolysis specificity of SpoIVA, we first substituted different residues in each motif, either individually or in several combinations. Next, we overproduced and purified the variants

from *E. coli* and tested the efficiency of each variant in hydrolyzing ATP and GTP in vitro. As a control, we compared these activities to that of *B. subtilis* Era GTPase that we purified using a similar protocol. We measured nucleotide hydrolysis for each variant at increasing nucleotide concentrations to produce saturation curves that revealed the substrate turnover rate ($k_{cat}$) and nucleotide concentration that produced half-maximal enzymatic activity ($K_m$) (*Figure 2—figure supplement 1*, *Supplementary file 2*). We then calculated the catalytic efficiency for each reaction ($k_{cat}/K_m$) which reflects how likely the forward reaction (hydrolysis of the bound nucleotide) will proceed (*Figure 2A*, *Supplementary file 2*). Wild-type (WT) SpoIVA displayed a catalytic efficiency of $4.2 \pm 0.9$ min$^{-1}$ mM$^{-1}$ for ATP, compared to just $1.3 \pm 0.8$ min$^{-1}$ mM$^{-1}$ for GTP, indicating that the protein hydrolyzed ATP approximately threefold more efficiently than GTP (*Figure 2B*). By comparison, Era did not display appreciable basal ATPase or GTPase activity, but upon incubation with an RNA oligonucleotide corresponding to the 16S rRNA sequence to which Era binds and which reportedly stimulates the enzymatic activity of Era (*Meier et al., 2000*; *Tu et al., 2011*), Era hydrolyzed GTP with a catalytic efficiency of $0.7 \pm 0.2$ min$^{-1}$ mM$^{-1}$, similar to the reported activity of *E. coli* Era (*Tu et al., 2011*). However, Era did not specifically hydrolyze ATP, as evidenced by the failure of the reaction to reach saturation and display Michaelis–Menten kinetics (*Figure 2—figure supplement 1J*, *Figure 2A*, 'I.D.' for 'indeterminable'). Restoring the Asp in the degenerate NKxD motif of SpoIVA did not significantly change the catalytic efficiencies of ATP or GTP hydrolysis but restoring either the Lys or full NKxD motif in SpoIVA resulted in a ~2.5-fold increase in the catalytic efficiency for ATP and ~3.5-fold of that for GTP hydrolysis. Restoring either the single Lys or the full NKxD motif resulted in decreased preference for ATP (*Figure 2B*). Curiously, these variants also displayed an unusually high turnover rate that was ~10-fold higher for both NTPs than that displayed by WT SpoIVA (*Figure 2—figure supplement 1C,D*).

We next investigated if altering the SxE sequence in SpoIVA would reduce the unusually high enzymatic activity resulting from restoration of the NKxD motif. Substituting both the Ser and Glu with Ala (to disrupt both positions with an amino acid with a short sidechain that is unlikely to perturb overall SpoIVA structure) in the context of the restored NKxD motif resulted in lowered catalytic efficiencies, similar to WT SpoIVA (*Figure 2A,C*: 'NKxD AxA'). This change also increased the catalytic efficiency of the enzyme for GTP, resulting in drastically reduced specificity for ATP relative to WT SpoIVA (*Figure 2B*). Changing the SxE sequence alone to AxA had a similar effect (*Figure 2A*, 'NSxR AxA'). Substituting the Glu alone in the SxE sequence with Ala ('NSxR SxA') mimicked the NSxR AxA variant with respect to nucleotide specificity (*Figure 2B*), but further lowered the catalytic efficiency of the enzyme (*Figure 2A*). Finally, combining a restoration of just the Asp residue of the degenerate NKxD motif with disruption of just the Glu of the SxE sequence, resulted in an enzyme that displayed a similar catalytic efficiency to that of Era (*Figure 2A*, 'NSxD SxA'; hereafter referred to as 'SpoIVA$^{GTPase}$'; *Figure 2C*) with a slight preference for GTP over ATP (*Figure 2B*).

To test if altering the G4 NKxD motif and introducing the SxE sequence in G5 were sufficient for the emergence of preferential ATPase activity, we changed the NKxD motif in Era to NSxR, to resemble the G4 sequence in SpoIVA, and substituted a glutamate at the end of the SAx sequence in Era to produce 'SAE' (thereby introducing an SxE motif) and tested the nucleotide hydrolysis activity of the evolved variant. Similar to WT Era, the evolved Era (Era$^{ATPase}$) did not exhibit a basal NTPase activity, but upon stimulation with the 16S rRNA fragment, Era$^{ATPase}$ hydrolyzed ATP with a catalytic efficiency of $0.2 \pm 0.1$ min$^{-1}$ mM$^{-1}$ (*Figure 2A*), but more curiously failed to specifically hydrolyze GTP (*Figure 2—figure supplement 1K*).

The key alterations in shifting the NTPase activity of SpoIVA toward that of Era were to restore the Asp of the NKxD and replace the Glu of the SxE (*Figure 2C*). Conversely, disrupting the NKxD motif of Era and introducing a Glu to create an SxE motif were sufficient to drive the preferential hydrolysis of ATP over GTP. The mutational analyses therefore indicate that the Asp of the NKxD motif contributes to nucleotide specificity and that the substitution of Asp to Arg seen in the extant *B. subtilis* SpoIVA contributes to altering the nucleotide-binding pocket to discriminate against GTP in favor of ATP. This is consistent with reported crystal structures of TRAFAC GTPases with a bound GTP (*Knihtila et al., 2015*) that show that the Asp of the NKxD motif can interact with the base of the bound nucleotide. In addition, another report showed that changing the Asp to another residue can alter the nucleotide-binding preference of Ras GTPase from GTP to xanthine triphosphate (XTP) (*Kang et al., 1994*; *Weijland et al., 1994*; *Zhong et al., 1995*). The Glu in the SxE sequence further contributes to ATP hydrolysis specificity and likely stabilizes the binding of ATP over GTP.

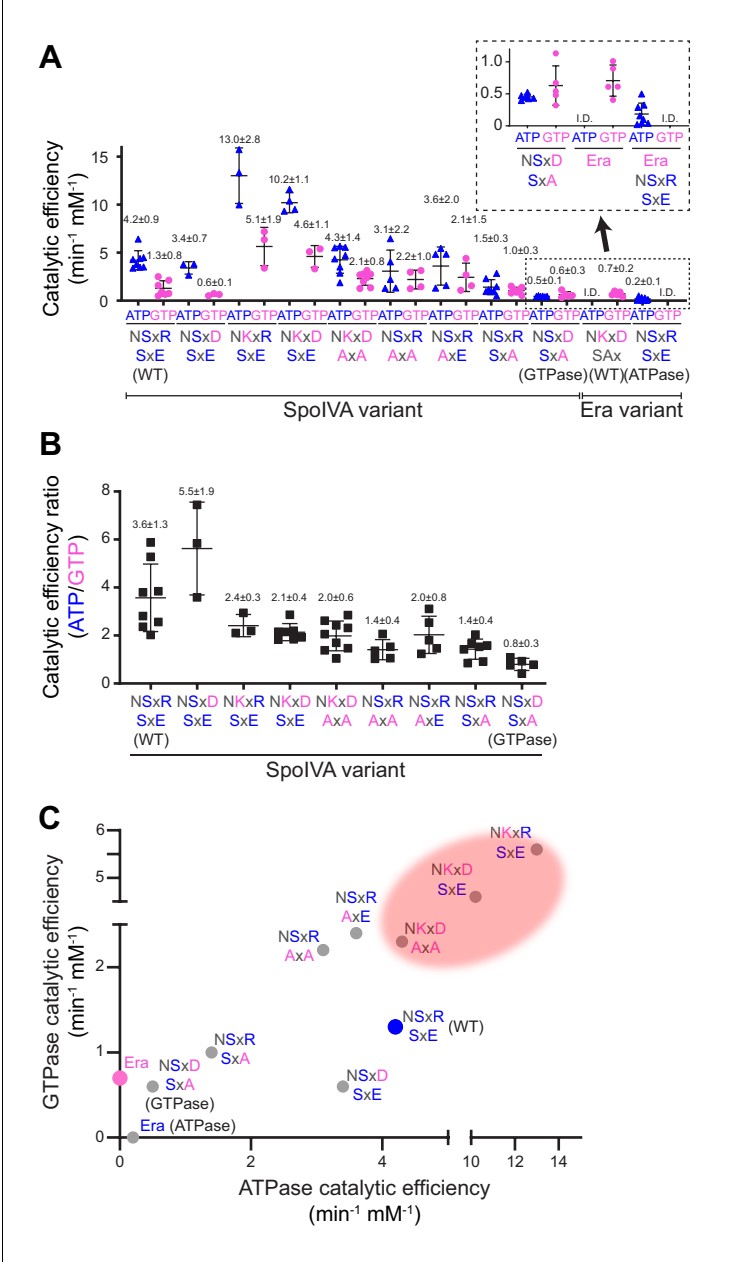

**Figure 2.** Stepwise restoration of ancestral GTPase activity in SpoIVA using site-directed mutagenesis. (**A**) Catalytic efficiencies of ATP (blue triangles) or GTP (pink circles) hydrolysis by different SpoIVA or Era variants, indicated by the amino acids substituted in the degenerate NKxD motif or the SxE motif. Catalytic efficiencies ($k_{cat}/K_m$) were calculated by measuring nucleotide hydrolysis for each SpoIVA or Era variant by increasing nucleotide concentration from 0 to 4 mM to produce saturation curves (*Figure 2—figure supplement 1*) that revealed the substrate turnover rate ($k_{cat}$) and nucleotide concentration that produced half-maximal enzymatic activity ($K_m$). $K_m$ and $k_{cat}$ values for each variant are reported in *Supplementary file 2*. Amino acids depicted in blue indicate that the residue was present in the extant (WT) SpoIVA ATPase; those depicted in pink indicate that the residue was altered to mimic the Era GTPase. Each data point represents mean results of an independent assay performed three to four times with one batch of purified protein; bars represent aggregate mean values from all experiments (also stated above each data set); error bars are S.D. Inset: magnification of data sets for the SpoIVA[GTPase] variant (NSxD, SxA) and Era variants. (**B**) Ratios of catalytic efficiencies for ATP and GTP hydrolysis by different SpoIVA variants. Data points represent ratios obtained from an independent parallel assay using ATP and GTP; bars represent mean values (also stated above each data set); error bars are S.D. (**C**) Catalytic efficiencies for GTP hydrolysis in (**A**) plotted as a function of ATP hydrolysis in (**A**) for each SpoIVA variant. Red shading indicates

*Figure 2 continued on next page*

*Figure 2 continued*

parameter space wherein SpoIVA variants are not functional in vivo (as reported in *Figure 3*, *Figure 3—figure supplement 1A*).

The online version of this article includes the following source data and figure supplement(s) for figure 2:

**Source data 1.** Raw data for enzyme kinetics.

**Figure supplement 1.** Saturation curves for ATP and GTP hydrolysis by SpoIVA and Era variants.

## Nucleotide promiscuity does not abrogate function of SpoIVA$^{GTPase}$ in vivo

To ensure that the amino acid substitutions introduced to restore ancestral SpoIVA GTPase activity did not completely abrogate protein function, we tested the ability of the different variants to complement the sporulation defect caused by a deletion of the *spoIVA* gene in *B. subtilis*. Deletion of *spoIVA* resulted in a >$10^8$-fold and a ~$10^6$-fold reduction in the production of heat resistant spores and lysozyme-resistant spores, respectively (*Roels et al., 1992*, *Figures 3A* and *Figure 3—figure supplement 1A*, *Supplementary file 2*), which could be complemented in trans by the introduction of WT *spoIVA* at an ectopic chromosomal locus. In contrast, while expression of the NSxD variant (which displayed similar enzymatic activity as WT SpoIVA in vitro; *Figures 2A* and *Figure 2—figure supplement 1B*, *Supplementary file 2*) complemented the *spoIVA* deletion, complementation by the hyperactive NKxR or NKxD variants resulted in >$10^8$-fold and ~$10^4$-fold decreases in the production of heat-resistant spores, respectively. Similarly, mutants that harbored a *spoIVA* allele containing a full substitution of the SxE sequence (resulting in AxA) also largely failed to sporulate when the G4 motif also harbored alterations (*Figures 3A* and, *Supplementary file 2*; NKxD AxA and NKxR AxA). Immunoblot analysis of extracts prepared from sporulating cells revealed that these SpoIVA variants were produced at levels similar to WT SpoIVA (*Figure 3—figure supplement 1B*). In an otherwise WT background, though, the AxA substitution (NSxR AxA) sporulated at near WT levels. Interestingly, SpoIVA$^{GTPase}$ (NSxD SxA), which showed reduced but similar hydrolysis of ATP and GTP (*Figure 2*), supported sporulation at near WT levels (*Figures 3A* and *Figure 3—figure supplement 1A*, *Supplementary file 2*). Although we cannot determine which nucleotide SpoIVA$^{GTPase}$ utilized in vivo, we can conclude that this disruption of the nucleotide-binding pocket did not result in either a large-scale structural defect in the protein that catastrophically affected its function or reduced its accumulation in vivo.

We next examined the subcellular localization in vivo during sporulation of each variant fused to green fluorescent protein, expressed from an ectopic chromosomal locus under control of the native *spoIVA* promoter. WT GFP-SpoIVA localized to the surface of the forespore (*Figure 3B,B'*), as did the NSxD variant which sporulated at near-WT levels (*Figure 3C,C'*). However, restoring only the Lys of the degenerate G4 motif or restoring the entire NKxD motif in an otherwise WT SpoIVA resulted in the mis-localization of the variant as a focus near the surface of the forespore. Ala substitution of the SxE sequence along with the NKxD motif resulted in a similar mis-localization pattern (*Figure 3D–F,D'–F'*). In contrast, various disruptions to the SxE sequence alone did not abrogate localization of the variant (*Figure 3G–I,G'–I'*). Finally, SpoIVA$^{GTPase}$ localized similar to WT (*Figure 3J–J'*), consistent with its ability to support sporulation at a near-WT level. Thus, disruption of the nucleotide-binding pocket of SpoIVA to permit the slightly preferential hydrolysis of GTP over ATP resulted in a protein that largely retained proper function in vivo.

## Hydrolysis of either ATP or GTP can drive a conformational change in SpoIVA

Although the sporulation efficiency and subcellular localization data indicated that SpoIVA$^{GTPase}$ was largely functional, it was difficult to infer if this variant used ATP or GTP to perform its function in vivo. We therefore monitored the nucleotide hydrolysis-driven conformational change and polymerization of SpoIVA in vitro in the presence of either nucleotide.

Structural changes in SpoIVA may be monitored by limited trypsin proteolysis (*Castaing et al., 2013*). We therefore incubated purified WT SpoIVA or variants with a low concentration of trypsin and assessed the extent of proteolysis at different time points by separating the reaction by Coomassie-stained SDS-PAGE. Importantly, the experiment was performed using 2 µM SpoIVA, which is

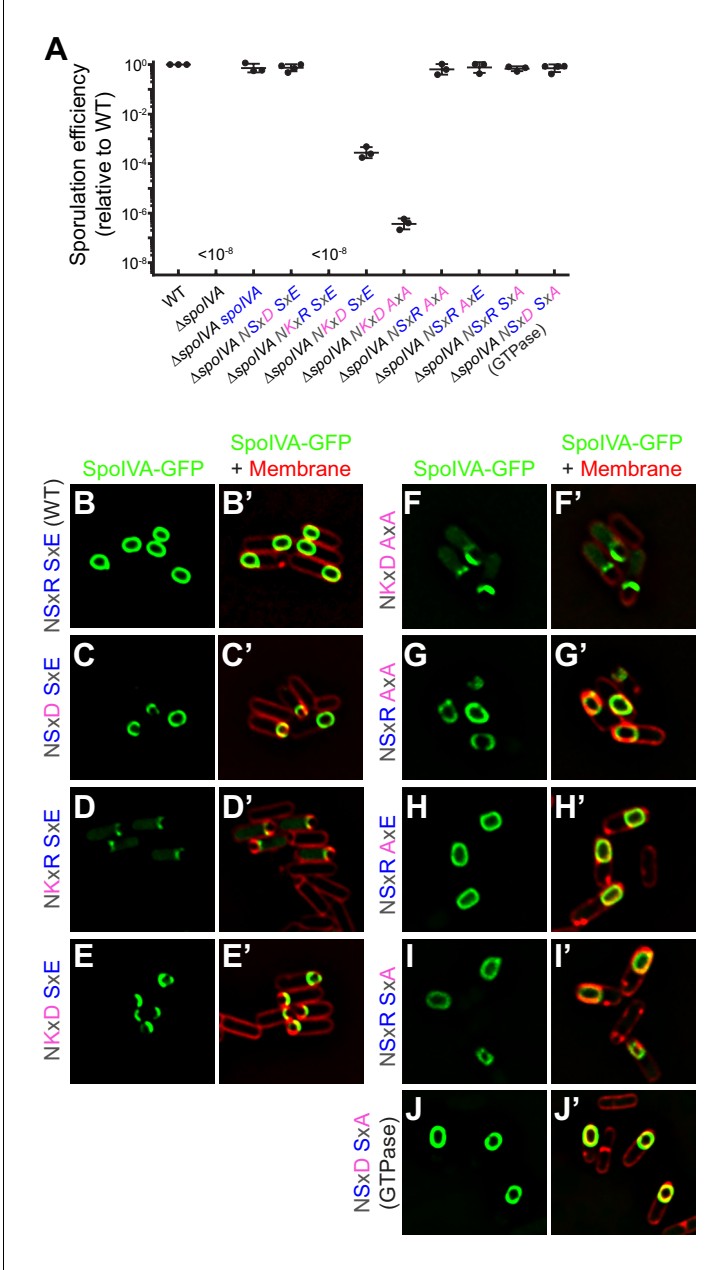

**Figure 3.** SpoIVA[GTPase] variant is functional in vivo. (**A**) Sporulation efficiencies, relative to WT (PY79) and measured as resistance to 80°C for 20 min, of *Bacillus subtilis* strains (PY79, KP73, KR394, NG7, NG13, NG8, TU209, TU211, TU212, TU213, and TU223) harboring the indicated allele of *spoIVA*. Data points represent sporulation efficiencies from independent cultures (n = 3–4); bars indicate mean values; error bars are S.D.; '<10$^{-8}$' indicates that no heat-resistant spores were recovered. Sporulation efficiencies are listed in *Supplementary file 2*. (**B–J**) Fluorescence micrographs of sporulating *B. subtilis* strains (SL55, JH19, JH20, JH21, TU200, TU201, TU202, TU203, and TU227) harboring the indicated SpoIVA variant fused to green fluorescent protein imaged 3 hr after the onset of sporulation. (**B–J**) Fluorescence from GFP; (**B'–J'**) overlay, GFP fluorescence from B to J , respectively, and fluorescence from membranes visualized using FM4-64. Genotypes are listed in *Supplementary file 1*. The online version of this article includes the following figure supplement(s) for figure 3:

**Figure supplement 1.** Lysozyme resistance and intracellular accumulation of SpoIVA and SpoIVA variants.

below the critical concentration for SpoIVA polymerization (*Figure 5—figure supplement 1A*; *Castaing et al., 2013*), and therefore reflective of polymerization-independent conformational changes in the protein. In the absence of nucleotide, SpoIVA was rapidly degraded, resulting in a characteristic banding pattern (*Figure 4A*). In contrast, co-incubation with either ATP or GTP resulted in a different banding pattern: most noticeably, the full-length protein was considerably resistant to degradation even after 10 min, suggestive of a massive conformational change in the protein upon hydrolysis of either nucleotide. The disappearance of full-length SpoIVA was quantified to produce a decay rate which indicated that a conformational change in WT SpoIVA could be achieved at an approximately similar rate by hydrolysis of either by ATP or GTP (*Figure 4B*). In contrast, restoring the Lys alone, the entire NKxD motif alone, or in combination with the entirely disrupted SxE sequence (to AxA) resulted in a rapid decay rate in the presence of either ATP or GTP, indicating that these variants were unable to achieve the characteristic nucleotide-dependent conformational change, consistent with the observed in vivo defects of these variants (*Figure 3*, *Supplementary file 2*). However, variants harboring substitution of either the Ser or Glu singly or together in the SxE sequence with Ala or SpoIVA$^{GTPase}$ displayed a decay rate in the presence of either nucleotide that was more similar to WT SpoIVA, suggesting that that these variants were able to utilize the energy released from hydrolysis of either ATP or GTP to drive the conformational change in the protein that is a prerequisite for polymerization (*Castaing et al., 2013*).

To further understand the nucleotide hydrolysis requirement for driving the conformational change in SpoIVA, we employed the limited trypsin digestion assay with SpoIVA and key SpoIVA variants using different nucleotides: ATP-γ-S and GTP-γ-S, which are non-hydrolyzable analogs of ATP and GTP; ADP and GDP; and ADP-AlF$_x$ and GDP-AlF$_x$, which are nucleotide analogs that mimic the transition state of ATP and GTP in the hydrolysis reaction (*Chen et al., 2007*; *Coleman and Sprang, 1999*). In the presence of ATP-γ-S or GTP-γ-S, WT SpoIVA displayed an intermediate conformational change suggesting that the protein bound, but did not hydrolyze, the nucleotide (*Figure 2—figure supplement 1L, M*), whereas SpoIVA harboring a Walker A disruption (SpoIVA$^{A*}$, which prevents ATP binding [*Ramamurthi and Losick, 2008*]) did not undergo a similar conformational change (*Figures 4C,D* and *Figure 4—figure supplement 1A-B*). Consistent with this result, the SpoIVA variant harboring a Sensor T disruption (SpoIVA$^{T*}$, which binds, but does not hydrolyze ATP [*Castaing et al., 2013*]) displayed a similar conformational change as WT SpoIVA in the presence of ATP-γ-S and GTP-γ-S (*Figures 4D* and *Figure 4—figure supplement 1C*), and SpoIVA$^{T*}$ binding to ATP. Together, this suggested that this intermediate conformational change is likely due to nucleotide binding and not due to slow hydrolysis of the bound nucleotide. Interestingly, the NKxD variant of SpoIVA, which exhibited elevated ATP and GTP hydrolysis levels (*Figure 2A*), displayed a conformational change in the presence of ATP-γ-S and GTP-γ-S similar to SpoIVA$^{A*}$ (*Figures 4D* and *Figure 4—figure supplement 1D*), suggesting that it did not stably undergo the initial conformational change that occurs upon binding the nucleotide. WT SpoIVA and SpoIVA$^{T*}$ exhibited a similar intermediate conformational change when incubated with ADP or GDP, but not the full conformational change that occurred upon nucleotide hydrolysis (*Figure 4E*). A similar intermediate conformational change upon binding GDP relative to binding GTP has been reported for other TRAFAC GTPases (*Coleman and Sprang, 1999*). Similar to the non-hydrolyzable nucleotides, SpoIVA$^{T*}$ and the NKxD variant did not display any conformational change when incubated with ADP or GDP (*Figure 4E*). Curiously, incubation of WT SpoIVA with either ADP-AlF$_x$ or GDP-AlF$_x$ (*Figure 4F*) produced a conformational change that was similar to that produced upon full hydrolysis of the nucleotide (*Figure 4B,C*). This suggested that the transition state mimics the activated state facilitated by ATP hydrolysis that is responsible for the conformational change seen in *Figure 4A*. As controls, incubation of the SpoIVA$^{T*}$, which is devoid of the alcoholic residue found to be critical for stabilizing the transition state of the protein-nucleotide complex in TRAFAC GTPases (*Coleman and Sprang, 1999*; *Leipe et al., 2002*), SpoIVA$^{A*}$, or the NKxD variant with ADP-AlF$_x$ or GDP-AlF$_x$ did not exhibit a similar conformational change (*Figure 4F*). Taken together, the results suggest that simply binding to ADP or GDP is insufficient to produce the full conformational change in SpoIVA required for polymerization. Instead, hydrolysis of an NTP molecule, while bound to the active site, is required for rearranging SpoIVA into a polymerization-competent state. The data are also consistent with a model in which rapid turnover of the bound NTP is incompatible with producing such a conformational change.

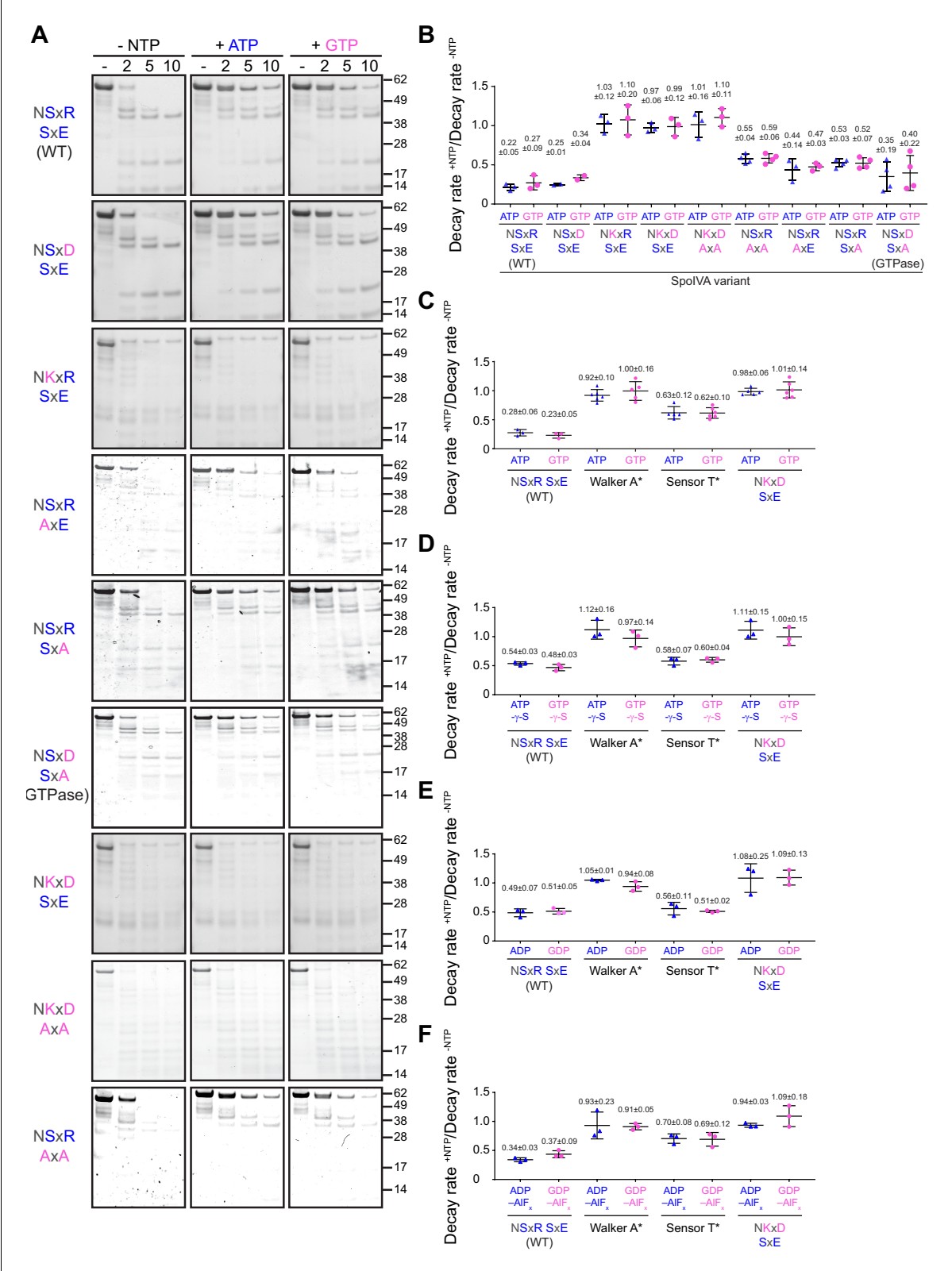

**Figure 4.** ATP or GTP hydrolysis, but not ADP or GDP binding, drives a conformational change in SpoIVA required for polymerization. (**A**) Purified variants of SpoIVA at 2 μM (below the threshold concentration for polymerization) were incubated either in the absence of nucleotide (left panels) or in the presence of ATP (middle) or GTP (right) at 37°C for 4 hr. Reactions were then exposed to limited proteolysis by trypsin for the indicated times (2, 5, or 10 min), after which proteolysis was stopped by addition of SDS sample buffer and the products were analyzed by Coomassie-stained PAGE.
*Figure 4 continued on next page*

*Figure 4 continued*

Mobility of molecular weight markers (kilodaltons) are indicated to the right. Displayed is a representative image (n = 3–4) (B) Quantification of the disappearance of the full length purified SpoIVA variants in (A) in the presence of ATP (blue triangles) or GTP (pink circles). Rates of decay are reported as a ratio of that in the presence to the absence of nucleotide (*Supplementary file 2*). (C–F) Quantification of the disappearance of the full length purified SpoIVA variant indicated (WT; Walker A* which does not bind ATP; Sensor T* which binds but does not hydrolyze ATP; NKxD SxE which hydrolyzes ATP at an increased rate) as in (B) in the presence of (C) ATP or GTP; (D) ATP-γ-S or GTP-γ-S; (E) ADP or GDP; or (F) ADP-AlF$_x$ or GDP-AlF$_x$. Representative images of Coomassie-stained gels for (C–F) are in *Figure 4—figure supplement 1*. Data points represent decay rate ratios from independent assays (n = 3–4); bars indicate mean values; error bars are S.D.

The online version of this article includes the following figure supplement(s) for figure 4:

**Figure supplement 1.** Representative SDS-PAGE images of limited trypsin proteolysis of purified SpoIVA and variants incubated with various nucleotides and nucleotide analogs.

## ATP, but not GTP, hydrolysis drives in vitro polymerization of SpoIVA$^{GTPase}$

Next, we tested the nucleotide-dependent polymerization rates of the SpoIVA variants by measuring the size distribution of polymerized SpoIVA molecules over time using dynamic light scattering (DLS). Incubation of purified WT SpoIVA, above the critical concentration for polymerization, with ATP, but not GTP, resulted in a steady increase in hydrodynamic radius (Rh) over a 5 hr period, consistent with ATP-dependent polymerization and what we previously observed (*Castaing et al., 2013*; *Figure 5—figure supplement 1A*). The initial slope of the polymerization reaction in the presence of nucleotide was quantified and reported relative to the initial slope of the reaction in the absence of nucleotide to yield polymerization rates for SpoIVA (*Figure 5A*, *Supplementary file 2*). This ratio revealed a greater than ~30% increase in polymerization rate with ATP than GTP (*Figure 5A*, *Supplementary file 2*), surprisingly indicating that the conformational change in SpoIVA driven by GTP hydrolysis (*Figure 4A,B,E*), which appeared similar to the conformational change driven by ATP hydrolysis, did not yield isomers of SpoIVA that were capable of polymerization.

We next examined the ability of SpoIVA variants in polymerizing with ATP and GTP. Restoring the Asp in the G4 motif, which did not display any obvious defect in the other assays, only slightly lowered the polymerization rate with ATP (*Figures 5A* and *Figure 5—figure supplement 1B*; *Supplementary file 2*). However, all variants that harbored a restoration of the Lys in the G4 motif were unable to polymerize with either nucleotide (*Figures 5A* and *Figure 5—figure supplement 1C-E*; *Supplementary file 2*), suggesting that substitutions leading to elevated nucleotide hydrolysis were unable to (1) induce a conformational change, (2) were defective in vivo, and (3) were also unable to promote SpoIVA polymerization. Disruptions to the SxE sequence lowered, but did not abolish, SpoIVA polymerization in the presence of ATP, but none of these variants polymerized in the presence of GTP (*Figure 5A*, *Figure 5—figure supplement 1F-H*; *Supplementary file 2*). Likewise, SpoIVA$^{GTPase}$, which was functional in vivo (*Figure 3*), and could use either ATP or GTP to induce a similar conformational change after nucleotide hydrolysis (*Figure 4A,B*), only polymerized in the presence of ATP (*Figures 5A* and *Figure 5—figure supplement 1I*; *Supplementary file 2*), suggesting a specific requirement for the nucleotide base, not simply the energy released from nucleotide hydrolysis, for its function.

## Formation of an ADP-bound SpoIVA multimeric intermediate is required for polymerization

Reported crystal structures of the eukaryotic septin GTPases, which also belong to the TRAFAC class of P-loop GTPases, show a dimer in which each monomer binds to a molecule of GDP that is stabilized by contacts from the other monomer with the guanosine base (*Zeraik et al., 2014*). Fully polymerized SpoIVA is devoid of any bound nucleotide (*Castaing et al., 2013*), but we wondered if SpoIVA would form an intermediate multimeric complex en route to polymerization whose formation would be dependent specifically on ADP binding before the hydrolyzed nucleotide was released. To test this, we first incubated 2 μM purified SpoIVA, below the threshold concentration for polymerization (*Figure 5—figure supplement 1A*), either in the absence of nucleotide or in the presence of ATP or GTP, and then separated the products by size exclusion chromatography (SEC). In the

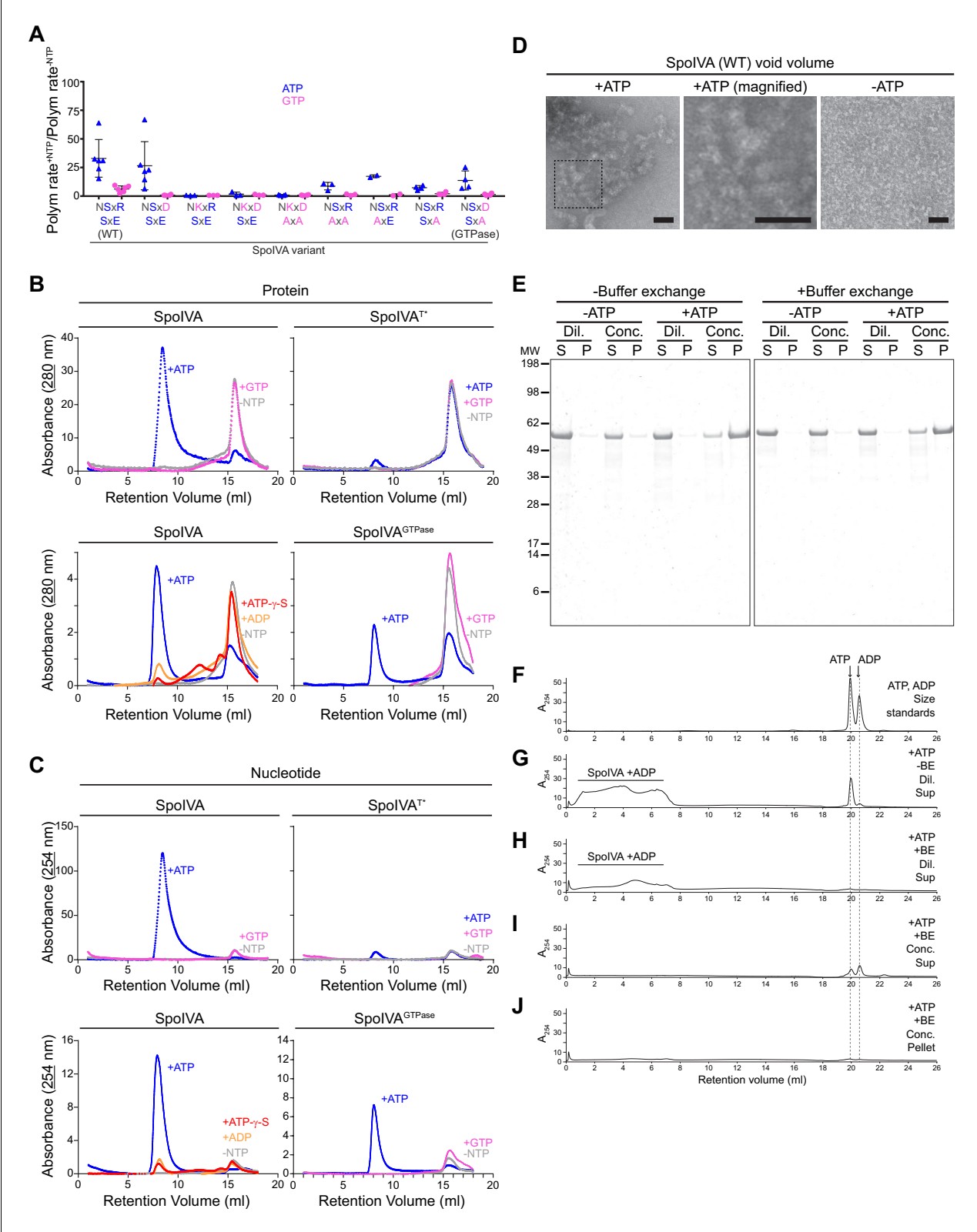

**Figure 5.** ATP, but not GTP, hydrolysis drives the formation of a functional assembly intermediate that is required for SpoIVA polymerization. (**A**) Initial polymerization rates of purified SpoIVA variants (6 µM) as measured by dynamic light scattering reported as a ratio of that in the presence and absence of the indicated nucleotide. Each data point represents a ratio obtained from an independent assay using ATP (blue triangles) or GTP (pink circles); bars represent mean values; error bars are S.D. Polymerization traces are in *Figure 5—figure supplement 1* and calculated rates are in

*Figure 5 continued on next page*

*Figure 5 continued*

***Supplementary file 2***. (B) Elution profiles of purified WT SpoIVA (top left), SpoIVA$^{T*}$ variant (which binds, but does not hydrolyze, nucleotide; top right [*Castaing et al., 2013*]), or SpoIVA$^{GTPase}$ (bottom right) that was incubated in the absence of nucleotide (gray), or presence of ATP (blue) or GTP (pink); or WT SpoIVA (bottom left) incubated with ATP-g-S (red) or ADP (orange); and separated by size exclusion chromatography (SEC) and detected using UV light absorbance at 280 nm (which measures aromatic rings in proteins). (C) Elution profiles of the identical experiments in (B) detected using UV light absorbance at 254 nm (which measures nucleotides). Depicted is a single representative experiment that was performed three times. (D) Negative stain transmission electron micrograph of the void volume obtained from SEC of WT SpoIVA in (B) incubated in the presence (left; indicated area shown at higher magnification in center panel) or absence (right) of ATP. Size bars: 50 nm. (E–J) SpoIVA assembly intermediate is functional for polymerization. Purified WT SpoIVA at 2 μM (below the threshold concentration for polymerization; *Figure 5—figure supplement 1A*) was incubated in the absence or presence of ATP at 37°C for 4 hr. Samples were divided in half and one half was buffer exchanged ('BE') to remove free ATP. Samples were then concentrated 20-fold to induce polymerization. Concentrated ('Conc.') and dilute ('Dil.') samples were then ultracentrifuged to collect polymerized material. (E) Supernatant (S) and resuspended pellet (P) fractions were separated by SDS-PAGE, SpoIVA was detected by Coomassie stain. Relative migration of molecular weight size markers (MW) is indicated to the left. (F–J) Indicated fractions were also separated by size exclusion chromatography and eluted material was detected using UV light absorbance at 254 nm. (F) Migration of ATP and ADP, as indicated. Supernatant fraction of purified SpoIVA incubated with ATP (G) without or (H) with buffer exchange ('BE'). Elution of SpoIVA bound to ADP in the column void volume is indicated. (I) Supernatant and (J) pellet fractions of purified SpoIVA incubated with ATP, after buffer exchange and concentration to induce polymerization, followed by heat denaturation to extract bound nucleotides (insoluble material was removed by centrifugation prior to loading the column).

The online version of this article includes the following source data and figure supplement(s) for figure 5:

**Source data 1.** Ion counts for intracellular nucleotide levels.
**Figure supplement 1.** Polymerization kinetics of SpoIVA and SpoIVA variants.
**Figure supplement 2.** Molecular weight determination of minimal the SpoIVA unit and assembly intermediate.
**Figure supplement 3.** ADP remains bound to SpoIVA assembly intermediate complex.

absence of nucleotide or the presence of GTP, SpoIVA migrated as a single peak (*Figure 5B*, top left; pink and gray traces). Quantitative mass imaging of this peak by employing interferometric scattering mass spectrometry (iSCAMS) (*Young et al., 2018*) revealed a mass of 103 ± 12 kDa (*Figure 5—figure supplement 2A*), similar to a predicted mass for a SpoIVA dimer of 114 kDa. Diluting the peak into a low salt buffer revealed an additional peak at 59 ± 8 kDa, similar to the predicted mass for a SpoIVA monomer of 57 kDa, along with the presumed dimeric peak of 109 ± 14 kDa corresponding to a SpoIVA dimer (*Figure 5—figure supplement 2B*). Addition of ATP resulted in shifting most of the protein to the void volume of the SEC column, indicating the formation of a larger complex (*Figure 5B*, top left; blue trace), even though the experiment was performed using a SpoIVA concentration that was below its critical concentration for polymerization (*Figure 5—figure supplement 1A*). In contrast, neither SpoIVA$^{T*}$ in the presence ATP or GTP, nor WT SpoIVA in the presence of either ATP-γ-S or ADP, formed a larger complex, indicating that ATP, but not GTP, hydrolysis is required for producing the species in the void volume (*Figure 5B*, top right and bottom left). Interestingly, incubation of SpoIVA$^{GTPase}$ with ATP, but not GTP, partially shifted a population into the void volume (*Figure 5B*, bottom right), indicating that the reengineered SpoIVA$^{GTPase}$ retained its specific dependence on ATP instead of GTP in forming the polymerization-competent intermediate, despite being able to hydrolyze both NTPs. Since the mass of the void volume peak was varied and too large to examine using iSCAMS, we employed size exclusion with multi-angle light scattering (SEC-MALS) analysis (*Some et al., 2019*), which revealed a range of molecular weights ranging from $2 \times 10^3$ kDa to $10^6$ kDa, suggesting multimers containing at least ~36 monomers of SpoIVA (*Figure 5—figure supplement 2C*). Examination of this void volume by negative stain and transmission electron microscopy revealed a species with a distinct structure that extensively self-interacted (*Figure 5D*). To check if this larger SpoIVA species may harbor an associated nucleotide, we examined the same peaks that eluted from the SEC column using 254 nm wavelength. Incubating WT SpoIVA or SpoIVA$^{GTPase}$ with GTP did not reveal a significant absorbance at 254 nm for fractions containing protein (*Figure 5C*, top left and bottom right; pink traces), but the protein peaks of the sample in the void volume when incubated with ATP displayed significant absorbance at 254 nm (*Figure 5C*, blue traces), suggesting the presence of nucleotide in this fraction. Neither SpoIVA$^{T*}$ incubated with ATP nor WT SpoIVA incubated with ATP-γ-S or ADP displayed significant absorbance at 254 nm (*Figure 5C*, top right and bottom left), suggesting that the nucleotide present in the void volume multimer of WT SpoIVA is likely ADP retained in the active site after hydrolysis. To confirm this, we extracted the nucleotide from the void volume fraction by denaturing

the protein. Separation of the extracted material by SEC revealed that it eluted at a similar volume as ADP, not ATP (*Figure 5—figure supplement 3*).

We next tested if the SpoIVA-ADP complex in the void volume observed in *Figure 5B* is a functional intermediate that can subsequently polymerize once its concentration exceeds the threshold concentration for polymerization. We therefore first incubated purified SpoIVA at low concentration (2 µM, below the threshold concentration for polymerization; *Figure 5—figure supplement 1A*) in the presence and absence of ATP. Half of each sample was then subjected to buffer exchange by SEC to remove free nucleotide. The desalted protein was then concentrated 20-fold using pressure dialysis after which polymerization was assayed by the formation of insoluble SpoIVA in the pellet fraction after ultracentrifugation (*Figure 5E*). In parallel, select supernatant and pellet fractions were separated by SEC to detect ATP and ADP (*Figure 5F–J*). Only samples that were incubated with ATP and whose concentration was increased displayed appreciable SpoIVA in the pellet fraction (*Figure 5E*). Importantly, removing free ATP prior to concentrating the sample also resulted in SpoIVA polymerization, suggesting that pre-incubation with ATP produces a functional ADP-bound SpoIVA multimeric intermediate that can polymerize once its threshold concentration for polymerization is subsequently achieved. Interestingly, while the protein-containing pellet fraction did not contain any bound nucleotide (*Figure 5J*; *Castaing et al., 2013*), the supernatant fraction contained ATP and ADP (*Figure 5I*), likely from ATP-bound SpoIVA that had not polymerized and consistent with the release of ADP upon SpoIVA polymerization.

Taken together, the results are consistent with a model in which SpoIVA, at a low concentration that does not promote polymerization, hydrolyzes ATP to undergo a conformational change and subsequently assembles into heterogeneous high molecular weight multimers that retain the ADP product of hydrolysis. Upon an increase in concentration, this functional SpoIVA multimeric intermediate releases the bound ADP and forms a nucleotide-free mature polymer. In contrast, while hydrolysis of GTP drove a conformational change in SpoIVA similar to what was achieved with ATP hydrolysis (*Figure 4A,B*), GTP hydrolysis did not permit formation of the high molecular weight intermediate (and therefore did not permit polymerization), nor did the protein retain GDP after hydrolysis (*Figure 5B,C*).

## Extant SpoIVA, but not SpoIVA^GTPase^, polymerizes in the presence of limiting level of ATP

As sporulation proceeds, intracellular ATP levels 2 hr after the induction of sporulation were reported to reach a high of ~1.5 mM while the level of GTP drops to a low of <0.06 mM (*Lopez et al., 1981*; *Lopez et al., 1979*; *Ochi et al., 1982*; *Ochi et al., 1981*). This relative abundance of ATP could therefore explain the evolutionary pressure that drove the switch in nucleotide-binding preference from GTP to ATP in SpoIVA. However, the time point at which these measurements were performed is before SpoIVA exerts its function during sporulation. We therefore harvested sporulating cells at various time points by vacuum filtration, extracted total nucleotides using organic solvent, and employed liquid chromatography-mass spectrometry (LC-MS) to quantify the relative abundance of individual nucleotides (*Figure 6A–D*). Immediately after induction of sporulation, ATP levels remained relatively constant (*Figure 6A*; blue trace, compare 'pre-induction' to 0 hr; *Supplementary file 3*), but increased almost twofold by the first hour, before returning to pre-sporulation levels in the second hour. At t = 3.5 hr, when SpoIVA is actively assembling the spore coat basement layer (*Peluso et al., 2019*; *Price and Losick, 1999*), ATP levels were ~70% of pre-sporulation levels; by t = 5 hr, ATP levels were less than 35% of pre-sporulation levels. Assuming a *B. subtilis* cell volume of 2.38 fL and using calculated LC-MS detection efficiencies for ATP and GTP (*Fung et al., 2020*), this corresponds to a pre-induction intracellular concentration of ATP of 2.3 mM ±0.89 mM in casein hydrolysate media; the concentration of ATP 3.5 hr after the induction of sporulation corresponds to 1.6 mM ± 0.25 mM, and 0.81 mM ± 0.084 mM at t = 5 hr, after achieving a concentration of 4.5 mM ± 0.044 mM at t = 1 hr. In contrast, GTP, which was present initially at 0.65 mM ± 0.33 mM plummeted to 0.071 mM ±0.034 mM immediately upon induction of sporulation (*Figure 6A*, pink trace; *Supplementary file 3*). Over the next 3.5 hr, GTP levels rose approximately threefold, and ended up twofold higher than at t = 0 after 5 hr. This corresponded to an approximately eightfold excess of ATP over GTP at t = 3.5 hr (intracellular concentration of 0.20 mM ±0.034 mM GTP), and an approximately fivefold excess of ATP at t = 5 hr (intracellular concentration of 0.16 mM ±0.012 GTP). The relative amounts of CTP and UTP were lower than that of ATP

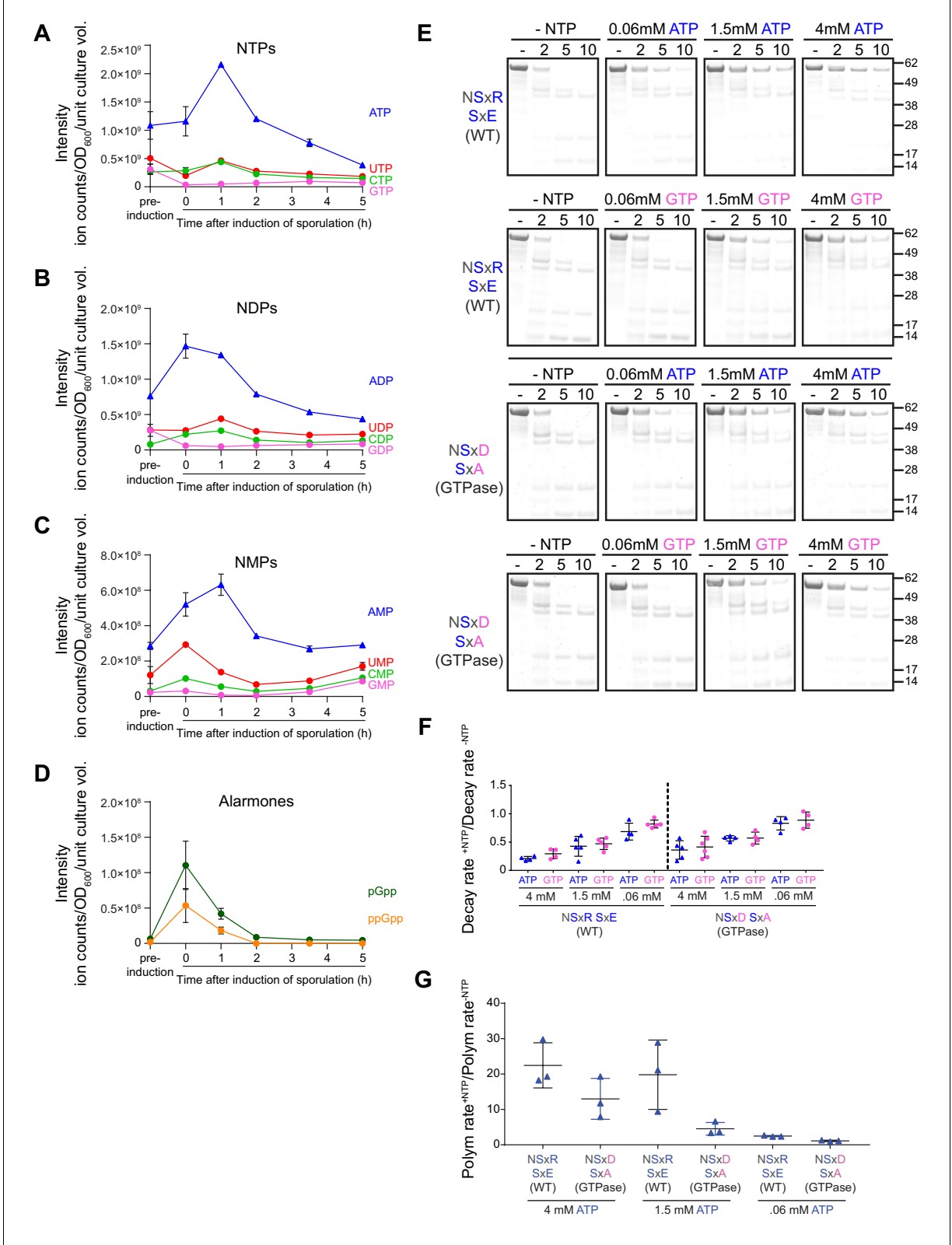

**Figure 6.** Extant SpoIVA polymerizes more efficiently than the SpoIVA[GTPase] in the presence of ATP. (A–D) Extraction of nucleotides from sporulating *B. subtilis* cultures at various time points and quantification using LC-MS. Quantification of (A) nucleoside triphosphates ATP (blue), GTP (pink), CTP (green), and UTP (red); (B) nucleoside diphosphates ADP (blue), GDP (pink), CDP (green), and UDP (red); (C) nucleoside monophosphates AMP (blue), GMP (pink), CMP (green), and UMP (red); and (D) alarmones ppGpp (orange) and pGpp (green). 'Pre-induction' indicates time point immediately prior

*Figure 6 continued on next page*

*Figure 6 continued*

to induction of sporulation; 0 hr is defined as immediately after sporulation induction. Data points indicate mean (n = 3 independent cultures); error bars are S.E.M. Ion count values are listed in *Supplementary file 3*. (E) Purified SpoIVA (top two rows) or SpoIVA$^{GTPase}$ (bottom two rows) were incubated with increasing concentrations of either ATP (rows 1 and 3) or GTP (rows 2 and 4) at 37°C for 4 hr and subjected to limited trypsin proteolysis for various lengths of time indicated, and the resulting products were analyzed by Coomassie-stained PAGE as described in *Figure 4A*. Mobility of molecular weight markers (kilodaltons) are indicated to the right. Displayed is a representative experiment (n = 3–5). (F) Quantification of the disappearance of the full length purified SpoIVA variants in (E) in the presence of ATP (blue triangles) or GTP (pink circles). Rates of decay are reported as a ratio of that in the presence over the absence of nucleotide. Each point represents an independent experiment (n = 3–5). (G) Initial polymerization rates of purified SpoIVA variants (6 µM) as measured by dynamic light scattering reported as a ratio of that in the presence (4 mM, 1.5 mM, or 0.06 mM ATP) over the absence of ATP. Each data point represents a ratio obtained from independent assays (n = 3) in the presence and absence of ATP; bars represent mean values; error bars are S.D.

during the first 5 hr of sporulation (*Figure 6A*), but slightly higher than that of GTP. Consistent with the increase in ATP level in the first hour of sporulation, the levels of ADP and AMP increased approximately twofold immediately upon induction of sporulation (*Figure 6B,C*, *Supplementary file 3*). Curiously, not only did the nucleotide alarmone ppGpp increase ~30-fold immediately upon induction of sporulation (*Figure 6D*, orange trace; *Supplementary file 3*) to 0.11 mM ± 0.088, similar to what was previously reported (*Ochi et al., 1982*), but the newly discovered guanosine nucleotide alarmone pGpp, which is produced by hydrolysis of (p)ppGpp (*Yang et al., 2020*), also increased ~30-fold immediately (0.19 mM ± 0.10 mM at t = 0 hr) after sporulation was induced.

Since SpoIVA$^{GTPase}$ functioned similar to the extant (WT) SpoIVA in vivo with respect to sporulation efficiency and localization (*Figures 3* and *Figure 3—figure supplement 1A*; *Supplementary file 2*), we wondered if the extant SpoIVA evolved to more efficiently utilize ATP at physiological nucleotide concentrations, in a way that was not evident by measuring sporulation efficiency by heat and lysozyme resistance (*Figures 3A* and *Figure 3—figure supplement 1A*). Both SpoIVA and SpoIVA$^{GTPase}$ failed to undergo nucleotide hydrolysis-mediated structural changes in the presence of 0.06 mM GTP at sub-polymerization levels of the protein but did so in the presence of 1.5 mM ATP (*Figure 6E,F*). However, when we examined SpoIVA polymerization at different concentrations of ATP, we observed that while neither SpoIVA nor SpoIVA$^{GTPase}$ were able to polymerize in the presence of 0.06 mM ATP (*Figure 6G*), WT SpoIVA, but not SpoIVA$^{GTPase}$, was able to efficiently polymerize in the presence of an intermediate concentration of ATP (1.5 mM; *Figure 6G*), similar to what was observed for WT SpoIVA using 4 mM ATP. Thus, although SpoIVA$^{GTPase}$ promiscuously hydrolyzed ATP and GTP (*Figure 2A*, *Figure 2—figure supplement 1J*) and could polymerize with excessive ATP (albeit at a slower rate; *Figures 5A* and *6G*), it was unable to polymerize under limiting concentration of ATP (*Figure 6G*). The limiting amounts of ATP (1.5 mM, the approximate amount of intracellular ATP between 3.5 hr and 5 hr of sporulation) therefore suggests a selective pressure that could have driven the initial amino acid substitutions required to switch nucleotide preference from GTP to ATP in SpoIVA and subsequent substitutions to enhance polymerization activity.

## Discussion

In this study we examined an unusual bacterial cytoskeletal protein, SpoIVA, that hydrolyzes ATP to drive the formation of static polymers. We previously reported that the SpoIVA ATPase is ancestrally derived (*Castaing et al., 2013*) from an Era-like GTPase, a pan-bacterial ribosomal maturation protein belonging to the TRAFAC class of P-loop GTPases (*Leipe et al., 2002*). We proposed that this likely occurred via a gene duplication event followed by a rapid divergence from the ancestral gene (*Castaing et al., 2013*). To understand the selective pressure underlying the switch in nucleotide specificity of the extant SpoIVA and the mechanistic details governing its function, we sought to restore its ancestral enzymatic (GTPase) activity by re-engineering its nucleotide-binding pocket and examining how the altered protein functioned in vivo and in vitro. Achieving this required altering amino acids in two loops near the base of the bound NTP (*Figure 1*). First, we partially restored the highly conserved NKxD motif on the G4 loop that has been implicated in conferring GTP-binding specificity (*Dever et al., 1987*) and is highly conserved among GTPases (*Leipe et al., 2002*), but is altered in SpoIVA. Second, we altered the sequence (SxE in *B. subtilis* SpoIVA) in the G5 loop that is

less conserved among GTPases (SAx). This approach resulted in a protein whose enzymatic activity operated in a parameter space similar to that of the ancestral Era GTPase and hydrolyzed GTP with a slight preference over ATP in vitro (*Figure 2—figure supplement 2 and 1*). Additionally, the altered protein was able to exploit the energy released by hydrolysis of either ATP or GTP to drive a critical conformational change required for polymerization (*Figure 4A,B*). Despite the ability of this protein to undergo an initial conformational change upon hydrolyzing either nucleotide, the protein only polymerized in the presence of ATP (*Figure 5A*), suggesting that the nucleotide base, and not just the energy released from nucleotide hydrolysis, was required for protein function.

This requirement for ATP led us to propose that the scarcity of GTP during the late stages of sporulation (*Figure 6A*) could have helped drive the evolution of SpoIVA to preferentially utilize ATP. Indeed, amino acid starvation and the onset of stationary phase are known to result in a reduction in GTP and GDP levels, which coincides with an increase in the production of the nucleotide alarmones (p)ppGpp (*Kriel et al., 2012*; *Liu et al., 2015*; *Ochi et al., 1981*). This drop in GTP level has also been implicated in the initiation of sporulation in *B. subtilis* (*Lopez et al., 1979*; *Ochi et al., 1982*), possibly through derepressing the activity of the CodY transcription factor that represses sporulation initiation genes by directly sensing GTP via its ligand-binding GAF domain (*Aravind and Ponting, 1997*; *Brinsmade, 2017*; *Sonenshein, 2005*). Here, we showed that GTP and GDP scarcity continued even 3.5–5 h after the initiation of sporulation, when SpoIVA is actively assembling the spore coat (*Peluso et al., 2019*), which suggests a selective pressure that drove the switch in nucleotide specificity in SpoIVA from GTP to ATP. Consistent with this notion, we observed that physiological levels of intracellular ATP, but not GTP, facilitate the requisite conformational changes in SpoIVA and SpoIVA$^{GTPase}$ (*Figure 6E,F*). This pressure caused by low levels of GTP is likely constrained by the fact that (p)ppGpp actively inhibits GTP production; in fact, artificially elevating GTP levels during nutrient limitation was shown to be detrimental to the cell (*Kriel et al., 2012*). Thus, the apparent requirement for low GTP levels during sporulation likely drove SpoIVA to utilize the more abundant ATP, rather than force the cell to generate more GTP.

One puzzling observation was that SpoIVA$^{GTPase}$, which promiscuously hydrolyzed ATP and GTP, functioned similar to the extant SpoIVA in vivo (*Figure 3*), which led us to wonder why the protein needed to have evolved further. However, when we employed a more sensitive assay which monitored the kinetics of SpoIVA polymerization in vitro, we found that, at an intermediate concentration of ATP that resembles the in vivo intracellular concentration of ATP during the late stages of sporulation (*Figure 6A*), the extant SpoIVA polymerized more robustly than did SpoIVA$^{GTPase}$. Thus, after GTP levels have dropped at the end of the sporulation program, and when ATP also gradually depletes, the extant SpoIVA has apparently evolved to better utilize ATP to drive efficient polymerization. We can speculate that the evolution from hydrolyzing GTP to ATP likely started with changing the Lys in the NKxD motif of Era to Ser in the ancestral SpoIVA, since any combination of SpoIVA mutants tested in this study that retain the Lys was not functional in vivo (*Figure 3*) or in vitro (*Figure 4*), likely due to elevated nucleotide hydrolysis (*Figure 2* and *Figure 2—figure supplement 1*). Thus, this substitution appears to have modulated the enzyme activity, probably allowing the appropriate conformational changes to occur upon nucleotide binding. The next likely change was the loss of the Asp in the G4 NKxD motif, which was shown previously to confer GTP specificity to other TRAFAC GTPases (*Cool et al., 1999*), and which resulted in a slight increase in catalytic efficiency and preference for ATP (*Figure 2A,B*), yet did not abrogate SpoIVA function in vivo and in vitro (*Figures 3* and *4*). Finally, the substitutions in the G5 loop such as the emergence of a polar position two residues downstream of the serine (*Glu in B. subtilis*) in the G5 loop of the extant SpoIVA contributed to the higher catalytic efficiency and preference for ATP we observe in vitro (*Figure 2*) and the more efficient polymerization we observed at the lower ATP levels (*Figure 6G*). Consistent with this model we find this mutational route to yield the most direct stepwise progression of catalytic efficiency of NTP hydrolysis from Era to extant SpoIVA (*Figure 2C*).

Our studies also revealed a stable polymerization intermediate that could explain the specific functional dependence of SpoIVA on ATP. At a low concentration of SpoIVA, which did not permit polymerization, we observed that SpoIVA formed a heterogeneous population of high molecular weight multimers in the presence of ATP, and not GTP (*Figures 5B* and *Figure 5—figure supplement 2C*). Further experiments revealed that formation of these multimers required ATP hydrolysis and that, in the absence of polymerization, the multimers bound the hydrolyzed nucleotide (ADP) (*Figures 5C* and *Figure 5—figure supplement 3*) and were capable of polymerizing after

subsequent removal of free ATP (*Figure 5E–J*). Our working model for SpoIVA polymerization (*Figure 7*) proposes that ATP hydrolysis results in the formation of high molecular weight SpoIVA multimers that stably bind ADP (when SpoIVA is present below the critical concentration for polymerization).

Several GTPases of the septin-GIMAP-dynamin clade within the TRAFAC class form oligomeric or polymeric assemblies typically in the proximity of lipid membranes and are involved in several aspects of membrane dynamics (*Schwefel et al., 2010*). SpoIVA represents a further independent example of the emergence of such polymerization activity within the TRAFAC class. However, some aspects of its dynamic oligomerization into higher order structures specifically resemble certain members of the septin-GIMAP-dynamin clade of GTPases. In particular, the ADP-dependent multimerization of SpoIVA is reminiscent of the manner in which septins multimerize when bound to GDP (*Zeraik et al., 2014*). Since the final SpoIVA polymer is nucleotide-free (*Figure 5J*; *Castaing et al., 2013*), the model predicts that polymerization of SpoIVA multimers, which only occurs when the concentration of SpoIVA exceeds a threshold concentration, releases the bound ADP (*Figure 7*). The transient binding of ADP to a polymerization intermediate is consistent with our previous observation that while phosphate is rapidly released upon ATP hydrolysis, release of the resulting ADP is slightly delayed (*Castaing et al., 2013*). In contrast, although the extant SpoIVA can hydrolyze GTP (*Figures 2A* and *Figure 2—figure supplement 1A*), the residues required to retain the hydrolyzed nucleotide are presumably no longer present, resulting in GDP release, which precludes formation of the NDP-bound multimeric intermediate required for polymerization (*Figure 7B*). This situation in a static structural protein, where either nucleotide may be accommodated but only one nucleotide promotes full function, is reminiscent of the binding of GTP by the enzyme adenylate kinase, wherein GTP binding arrests the protein in a catalytically inhibited conformation, but ATP binding permits large structural changes in the enzyme required for catalysis (*Rogne et al., 2018*).

Multiple examples in biology feature GTP-binding proteins that most commonly exploit nucleotide binding and hydrolysis as a timer or switch to relay a signal, whereas ATP is usually used by proteins participating in energy-intensive processes to perform work (*Alberts, 2002*). Since SpoIVA

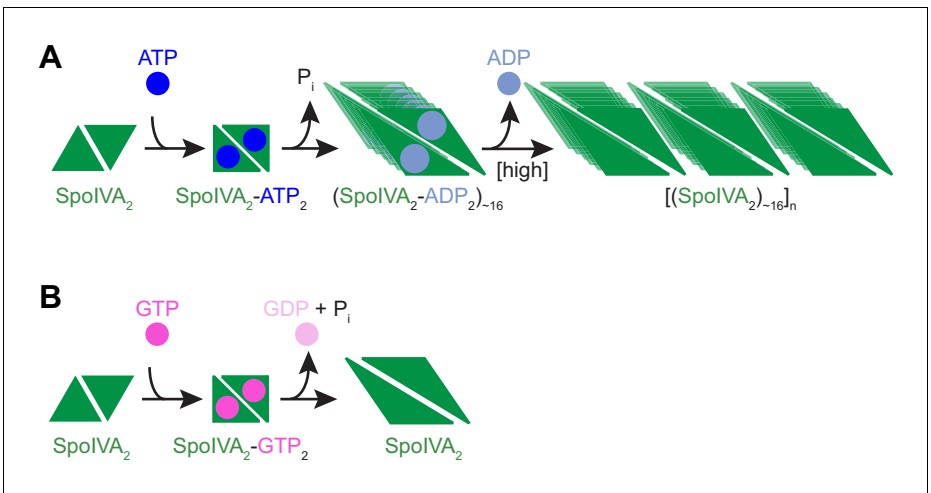

**Figure 7.** Model for the nucleotide-specific polymerization of SpoIVA. (**A**) Depicted is a SpoIVA dimer (green equilateral triangles; *Figure 5—figure supplement 2A,B*) that binds to ATP, resulting in a conformational change. Hydrolysis of the bound ATP (*Figure 2—figure supplement 1A*) drives a second conformational change in SpoIVA (*Figures 4A* and *6E*). The inorganic phosphate is released (*Castaing et al., 2013*), but the ADP remains bound temporarily (*Figure 5—figure supplement 3*), which we propose mediates multimerization of SpoIVA to form an assembly intermediate (*Figures 5B,C* and *Figure 5—figure supplement 2C*). At high enough concentration of SpoIVA, the ADP is released as SpoIVA multimers form static polymers (*Figure 5E–J*). (**B**) In the presence of high concentration of GTP, GTP hydrolysis by SpoIVA drives a conformational change in the protein similar to that observed in the presence of ATP (*Figure 4A*). However, GDP is prematurely released, SpoIVA fails to form the assembly intermediate (*Figure 5B,C*), and thus SpoIVA polymerization does not occur (*Figures 5A* and *Figure 5—figure supplement 1A*).

ultimately forms a static polymer and does not perform obvious work like a motor protein, the purpose of its ATP utilization had been mysterious. One implication of our model is that SpoIVA retained the ancestral 'switch' function of TRAFAC GTPases (nucleotide hydrolysis-dependent triggering of a conformational change). Indeed, it is likely that the precursor of SpoIVA was initially recruited for a structural role due to the capacity of GTPases to form nucleotide-dependent oligomeric assemblies in proximity to membranes as also observed in the GIMAP-septin-dynamin clade. However, as it became fixed for this function in the context of sporulation, SpoIVA appears to have substituted ATP for GTP as the molecule that mediates the switch because of the relative scarcity of GTP during the late stages of sporulation (*Figure 6A*; *Lopez et al., 1981*; *Lopez et al., 1979*; *Ochi et al., 1982*; *Ochi et al., 1981*). Functionally, this activity also resembles that of certain ATPases in the STAND clade of P-loop NTPases of the AAA+ class (*Leipe et al., 2004*). These NTPases, which include the apoptosis regulator Apaf-1 and the bacterial AfsR-like transcription regulators, employ ATP (and in some cases GTP) hydrolysis to transmit a conformational change to an effector domain to convey a signal, rather than perform a motor function (*Danot et al., 2009*; *Leipe et al., 2004*). In the future, detailed structural analyses of SpoIVA will likely yield insights into additional residues that evolved to increase the specificity of ATP binding and provide an atomic-scale mechanism for ATP-dependent multimerization and polymerization.

# Materials and methods

## Key resources table

| Reagent type (species) or resource | Designation | Source or reference | Identifiers | Additional information |
|---|---|---|---|---|
| Strain, strain background (*Bacillus subtilis*) | PY79 | *Youngman et al., 1984* | | Wild type |
| Strain, strain background (*Bacillus subtilis*) | KP73 | *Price and Losick, 1999* | | ΔspoIVA::neo |
| Strain, strain background (*Bacillus subtilis*) | KR394 | *Ramamurthi and Losick, 2008* | | ΔspoIVA::neo thrC::spoIVA spec |
| Strain, strain background (*Bacillus subtilis*) | NG7 | This paper, *Figure 3*, *Figure 3—figure supplement 1*, *Supplementary file 1* | | ΔspoIVA::neo thrC:: spoIVA$^{S189K}$ spec |
| Strain, strain background (*Bacillus subtilis*) | NG13 | This paper, *Figure 3*, *Figure 3—figure supplement 1*, *Supplementary file 1* | | ΔspoIVA::neo thrC:: spoIVA$^{R191D}$ spec |
| Strain, strain background (*Bacillus subtilis*) | NG8 | This paper, *Figure 3*, *Figure 3—figure supplement 1*, *Supplementary file 1* | | ΔspoIVA::neo thrC:: spoIVA$^{S189K, R191D}$ spec |
| Strain, strain background (*Bacillus subtilis*) | TU209 | This paper, *Figure 3*, *Figure 3—figure supplement 1*, *Supplementary file 1* | | ΔspoIVA::neo thrC:: spoIVA$^{S189K, R191D, S216A, E218A}$ spec |
| Strain, strain background (*Bacillus subtilis*) | TU210 | This paper, *Figure 3— figure supplement 1* | | ΔspoIVA::neo thrC:: spoIVA$^{S189K, S216A, E218A}$ spec |

*Continued on next page*

*Continued*

| Reagent type (species) or resource | Designation | Source or reference | Identifiers | Additional information |
|---|---|---|---|---|
| Strain, strain background (*Bacillus subtilis*) | TU211 | This paper, *Figure 3*, *Figure 3—figure supplement 1*, *Supplementary file 1* | | ΔspoIVA::neo thrC:: spoIVA$^{S216A, E218A}$ spec |
| Strain, strain background (*Bacillus subtilis*) | TU212 | This paper, *Figure 3*, *Figure 3—figure supplement 1*, *Supplementary file 1* | | ΔspoIVA::neo thrC:: spoIVA$^{S216A}$ spec |
| Strain, strain background (*Bacillus subtilis*) | TU213 | This paper, *Figure 3*, *Figure 3—figure supplement 1*, *Supplementary file 1* | | ΔspoIVA::neo thrC:: spoIVA$^{E218A}$ spec |
| Strain, strain background (*Bacillus subtilis*) | TU223 | This paper, *Figure 3*, *Figure 3—figure supplement 1*, *Supplementary file 1* | | ΔspoIVA::neo thrC:: spoIVA$^{R191D, E218A}$ spec |
| Strain, strain background (*Bacillus subtilis*) | SL55 | This paper, *Figure 3* *Supplementary file 1* | | ΔspoIVA::neo thrC::GFP-spoIVA spec ΔamyE::spoIVA cat |
| Strain, strain background (*Bacillus subtilis*) | JH19 | This paper, *Figure 3* *Supplementary file 1* | | ΔspoIVA::neo thrC::GFP-spoIVA$^{S189K}$ spec ΔamyE:: spoIVA$^{S189K}$ cat |
| Strain, strain background (*Bacillus subtilis*) | JH20 | This paper, *Figure 3* *Supplementary file 1* | | ΔspoIVA::neo thrC:: GFP-spoIVA$^{R191D}$ spec ΔamyE:: spoIVA$^{R191D}$ cat |
| Strain, strain background (*Bacillus subtilis*) | TU200 | This paper, *Figure 3* *Supplementary file 1* | | ΔspoIVA::neo thrC::GFP-spoIVA$^{S189K, R191D, S216A, E128A}$ spec ΔamyE::$^{spoIVAS189K, R191D, S216A, E218A}$ cat |
| Strain, strain background (*Bacillus subtilis*) | TU201 | This paper, *Figure 3* *Supplementary file 1* | | ΔspoIVA::neo thrC::GFP-spoIVA$^{S216A, E218A}$ spec ΔamyE:: spoIVA$^{S216A, E218A}$ cat |
| Strain, strain background (*Bacillus subtilis*) | TU202 | This paper, *Figure 3* *Supplementary file 1* | | ΔspoIVA::neo thrC::GFP-spoIVA$^{S216A}$ spec ΔamyE:: spoIVA$^{S216A}$ cat |
| Strain, strain background (*Bacillus subtilis*) | TU203 | This paper, *Figure 3* *Supplementary file 1* | | ΔspoIVA::neo thrC::GFP-spoIVA$^{E218A}$ spec ΔamyE:: spoIVA$^{E218A}$ cat |

*Continued on next page*

*Continued*

| Reagent type (species) or resource | Designation | Source or reference | Identifiers | Additional information |
|---|---|---|---|---|
| Strain, strain background (*Bacillus subtilis*) | TU227 | This paper, *Figure 3 Supplementary file 1* | | Δ*spoIVA::neo thrC::GFP-spoIVA*[R191D, E218A] *spec* Δ*amyE::spoIVA*[R191D, E218A] *cat* |
| Commercial assay or kit | Malachite Green Phosphate Assay Kit | BioAssay Systems | POMG-25H | |
| Antibody | Rabbit polyclonal anti-SpoIVA | Ramamurthi lab | | Raised against purified *B. subtilis* His$_6$-SpoIVA (1:20,000) |
| Antibody | Rabbit polyclonal anti-SigA | Ramamurthi lab | | Raised against purified *B. subtilis* SigA (1:50,000) |

## Sequence analysis

Starting sets of members of each GTPase family were collected by running BLASTP searches (*Altschul et al., 1997*; *Aravind and Koonin, 1999*) against a database of 4440 complete genomes and 2983 metagenomes (coding for a total of 21,646,808 proteins) obtained from the genomes division of Genbank (ftp://ftp.ncbi.nlm.nih.gov/genomes/). These were then filtered by similarity-based clustering with the BLASTCLUST program (https://www.ncbi.nlm.nih.gov/Web/Newsltr/Spring04/blastlab.html) to obtain the representative sets for each family. They were then aligned using the Kalign program (*Lassmann et al., 2009*; *Lassmann and Sonnhammer, 2005*) and further improved by examining GTPase structures. The multiple sequence alignments thus obtained were used to compute the sequence logos for each family. The logos were computed using the codebase obtained from RWeblogo (https://CRAN.R-project.org/package=RWebLogo) with the residue size scaled as per the probability of their occurrence in a column of the alignment. The NTPase active site pockets were drawn using the MarvinSketch program (https://chemaxon.com/products/marvin).

## Protein purification

His$_6$-tagged SpoIVA was overproduced in *E. coli* BL21(DE3) from plasmid pJP120 (WT SpoIVA) (*Castaing et al., 2013*) or pJP120 derivatives (SpoIVA variants constructed via QuikChange kit, Agilent) and purified using Ni$^{2+}$ affinity chromatography (Qiagen) and subsequently by ion-exchange chromatography (Mono Q; Pharmacia) as described previously (*Wu et al., 2015*). Briefly, 4 × 500 ml cultures of BL21(DE3) harboring pJP120 or derivatives [pIL48 (*spoIVA*[S189K]), pIL49 (*spoIVA*[R191D]), pIL50 (*spoIVA*[S189K,R191D]), pJH25 (*spoIVA*[S189K,R191D,E218A]), pJH26 (*spoIVA*[S189K,R191D,S216A]), pJH27 (*spoIVA*[S189K,R191D,S216A,E218A]), pTU141 (*spoIVA*[S216A,E218A]), pTU142 (*spoIVA*[S216A]), pTU143 (*spoIVA*[E218A]), and pTU220 (*spoIVA*[R191D,E218A])] were grown at 37°C in Terrific Broth (Fisher Scientific) containing 50 mg/ml kanamycin for plasmid maintenance to mid-logarithmic phase (~2.5 hr). Isopropyl-β-D-thiogalactopyranoside (Calbiochem, Millipore) was added to 1 mM final concentration to induce protein production and each culture was grown for 4 hr at 37°C. Harvested cells (which could be stored at −80°C) were resuspended in 25 ml of ice-cold Buffer A (50 mM Tris at pH 7.5, 150 mM NaCl) and disrupted by French Pressure Cell Press (SLM Aminco) at 12,000 psi. All subsequent steps were performed on ice. Unbroken cells and cell debris were removed by centrifugation at 35,000 rpm for 1 hr at 4°C and the cleared lysate was loaded on a single gravity column containing 3 ml of Ni-NTA agarose (QIAGEN), pre-equilibrated with ice-cold Buffer A, and incubated on ice for 30 min. Upon flow-through of the clarified lysate, the column was washed with 50 ml Wash Buffer I (Buffer A containing 20 mM imidazole), followed by 4 ml Wash Buffer II (Buffer A containing 80 mM imidazole). Protein was eluted with 10 ml ice-cold Elution buffer (Buffer A containing 250 mM imidazole). Imidazole was removed from eluted fractions using a PD-10 desalting column (GE Healthcare; 3.3 ml eluate/PD-10 desalting column) and eluted using 4 ml Buffer A. Peak fractions were identified using

NanoDrop $A_{280}$ (ND-1000, Thermo Scientific), pooled, separated by ion exchange chromatography (Mono Q 5/50, GE Healthcare), and then eluted with a step-wise gradient of 150–1000 mM NaCl; His$_6$-SpoIVA routinely eluted at 0.4 M NaCl. Purified protein was stored at 4°C and was used in less than 48 hr after purification due to precipitation of the protein upon prolonged storage. For long-term storage, samples were flash-frozen on dry ice and stored at −80°C. To assess the multimerization of SpoIVA and variants, 2 µM purified WT or T* SpoIVA variant (*Castaing et al., 2013*) was incubated with no NTP, 4 mM ATP, or 4 mM GTP in Buffer B (50 mM Tris at pH 7.5, 400 mM NaCl, 5 mM MgCl$_2$) for 4 hr at 37°C. 700 µl of each sample was then passed through a 0.22 µm filter and separated using a Superose 6 Increase 10/300 GL size exclusion column (GE Healthcare) with Buffer B at a flow rate of 0.2 ml/min. For assessing the presence of ATP and ADP, where indicated, samples were heated at 95°C for 20 min to release protein bound nucleotide, centrifuged to remove insoluble material, and the supernatant separated using Superdex 30 Increase 10/30 GL size exclusion column (GE Healthcare) with 50 mM Tris at pH 7.5 at a flow rate of 0.2 ml/min. Retention volumes were compared to that of free ATP and ADP standards (Sigma). *his$_6$-tagged era* gene was PCR-amplified using primers (5'-GGGGAATTGTGAGCGGATAACAATTC which abutted an *Xba*I restriction site, and 5'-GCTTGTCGACGGAGCTCGAATTCGGATCTTAATATTCGTCCTCTTTAAAGCCAAAATC which abutted a *BamH*I restriction site and six histidine codons) from *B. subtilis* PY79 chromosomal DNA and cloned into vector pET28a strain to generated plasmid pJH17C. His$_6$-tagged Era was over-produced in *E. coli* BL21(DE3) from plasmid pJH17C or pTU281 (harboring His$_6$-tagged Era$^{K125S, D127R, L156E}$ variant constructed via QuikChange site-directed mutagenesis kit, Agilent) and purified using Ni$^{2+}$ affinity chromatography (Qiagen) and subsequently by ion-exchange chromatography similar to purification of SpoIVA. However, since Era is positively charged at neutral pH, imidazole elutions were further separated by ion exchange chromatography using Mono S column (Mono S 5/50, GE Healthcare) instead of Mono Q. Purified protein was either stored at −80°C, after quick freeze on dry-ice, for long-term storage, or stored at 4°C and used within 48 hr.

## NTP hydrolysis

Different concentrations (ranging from 0 to 4 mM) of ATP or GTP (Sigma) were incubated with 0.3 µM purified His$_6$-SpoIVA or SpoIVA variants, or His$_6$-Era or Era variants in 50 µl Buffer B for 1 hr at 37°C. For reactions with Era and Era variants, 1 mM of the oligonucleotide rAUCACCUCCUUUCUA (corresponding to *B. subtilis* 3' end of the 16S rRNA) was added to the reaction prior to the addition of NTPs in order to stimulate Era hydrolysis (*Tu et al., 2011*). Concentration of released inorganic phosphate was determined using Malachite Green Phosphate Assay kit (BioAssay Systems) according to manufacturer's protocol. Briefly, reactions were stopped by the addition of 950 µl of water; 80 µl of diluted reaction was added to a single well of a flat-bottom 96-well plate (Costar). 20 µl of Malachite Green working reagent was added to each well and the reaction was incubated at room temperature for 30 min. Absorbance at 620 nm (Spark 10M plate reader, Tecan) of each reaction was compared to absorbances of known concentrations of phosphate standards. Absorbances from control reactions performed in the absence of SpoIVA for each NTP concentration were subtracted from absorbances of the respective reactions with SpoIVA to eliminate background hydrolysis. Hydrolysis rates for each NTP concentration were plotted using GraphPad Prism 7; $V_{max}$ and $K_m$ values were determined by fitting the data to Michaelis–Menten equation using best-fit values.

## Limited trypsin proteolysis

Limited proteolysis of His$_6$-SpoIVA and variants by partial trypsin digest was conducted as previously described (*Castaing et al., 2013*). Briefly, 2 µM His$_6$-SpoIVA was incubated in 100 µl of Buffer B supplemented with 4 mM NTP for 4 hr at 37°C. After addition of 1 µg/ml of trypsin (Sigma; diluted in 20 mM MgCl$_2$, 1 mM HCl), 15 µl of the reaction was removed and added at the indicated time points (0, 2, 5, and 10 min) to 5 µl of 4× LDS Sample Buffer (Invitrogen) containing beta-mercaptoethanol (Sigma) and heated at 95°C for 30 min to arrest proteolysis. 10 µl of each sample was separated by SDS-PAGE and stained with Coomassie blue. The intensity of the full-length His$_6$-SpoIVA band in each lane was quantified using ImageJ software (NIH), plotted as a function of time, and fitted to single-phase exponential decay using GraphPad Prism 7; reaction rates from each His$_6$-SpoIVA variant were normalized to the reaction rate of WT protein.

## In vitro polymerization
### Dynamic light scattering
6 µM purified His$_6$-SpoIVA in Buffer B (150 µl reaction volume) was incubated in the presence or absence of 4 mM NTP for 4 hr. At indicated time points, reactions were exposed to laser light in a DynaPro NanoStar System photometer (Wyatt Technology). Scattered light was measured as photons per second and analyzed using Dynamics V6 software (Novell) and the data were presented as hydrodynamic radius ($R_h$) and plotted in GraphPad (Prism 6) where initial polymerization rates were estimated using best-fit linear equations.

### Ultracentrifugation
To separate insoluble (polymerized) from soluble (non-polymerized) SpoIVA, 2 ml of 2 µM His$_6$-SpoIVA was incubated in Buffer B in the presence or absence of 4 mM ATP for 4 hr at 37°C. Half the sample was buffer exchanged to remove free ATP (Zeba Spin Desalting column, 7K MWCO, Thermo Fisher Scientific). Samples were then concentrated 20-fold (Amicon Ultra 3K MWCO, Millipore). Concentrated and non-concentrated samples (100 µl each) were centrifuged at 100,000 × g at 4°C for 30 min. The supernatant (95 µl) and the pellet (resuspended with 95 µl Buffer B) were collected, 15 µl of each was separated by SDS-PAGE gel, and visualized using Coomassie blue.

### Gel Filtration
For *Figure 5B,C*, 2 µM purified His$_6$-SpoIVA, His$_6$-SpoIVA$^{T*}$ or His$_6$-SpoIVA$^{GTPase}$ in Buffer B (1 ml reaction volume) was incubated in the presence of 4 mM ATP, GTP, ADP, or ATP-γ-S (Sigma) at 37°C for 4 hr. Reactions were centrifuged at 14,000 × g for 10 min to remove insoluble material and supernatant was separated on a Superose 6 Increase 10/300 GL column (GE Healthcare) at a flow rate of 0.25 ml/min. Chromatograms were generated by monitoring A$_{254}$ and A$_{280}$ as function of flow-through volume. For *Figure 5—figure supplement 3*, a Superdex 30 Increase 10/300 GL column (GE Healthcare) at a flow rate of 0.25 ml/min was used to separate ATP and ADP standards (Sigma) and material in the void volume of *Figure 5B,C*.

## Epifluorescence microscopy
Fluorescence microscopic images of WT and mutant *B. subtilis* were taken as previously described (*Ebmeier et al., 2012*). Briefly, overnight cultures of *B. subtilis* grown in casein hydrolysate (CH) media at 22°C were diluted 1:20 into 20 ml CH and grown at 37°C for 2 hr. Sporulation was induced via resuspension method (*Sterlini and Mandelstam, 1969*) in A+B media supplemented with 80 µg/ml threonine (Sigma) at 37°C. After 3.5 hr, cells were harvested and resuspended in PBS (KD Medical) containing 1 µg/ml FM4-64 (Invitrogen) to visualize membranes, then placed on lysine-coated glass bottom dish (MatTek Corp.) under a 1% agarose pad. Cells were viewed with a DeltaVision Core microscope system (Applied Precision) equipped with an environmental control chamber. Images were captured with a Photometrics CoolSnap HQ2 camera. Seventeen planes were acquired every 0.2 µm at 22°C, and the data were deconvolved using SoftWorx software (GE Healthcare). At the sporulation time points that we examined, phase bright forespores had not yet developed; thus, the autofluorescence of forespores was not higher than background fluorescence. Additionally, control experiments with sporulating strains that did not harbor a *gfp* fusion indicated that the level of GFP fluorescence from fusions to SpoIVA was well above the limited background fluorescence of the cells.

## Sporulation efficiency
To determine sporulation efficiencies, WT and mutant *B. subtilis* cells were grown in Difco Sporulation Medium for at least 24 hr at 37°C. Cultures were then exposed to 80°C for 20 min to kill non-sporulating cells. Surviving cells were enumerated by serial dilution and plating on LB agar. Viable spores were counted as colony forming units (CFUs); sporulation efficiencies were reported as a ratio to CFUs recovered from a parallel experiment using WT *B. subtilis*.

## Immunoblotting
Steady state levels of SpoIVA and variants were assessed via immunoblotting as previously described (*Tan et al., 2015*). Briefly, *B. subtilis* cells were induced to sporulate via resuspension as described

above. Sporulating cells were harvested and resuspended in 500 µl protoplast buffer (0.5 M sucrose, 10 mM $K_2PO_4$, 20 mM $MgCl_2$, and 0.1 mg/ml lysozyme [Sigma]) and incubated at 37°C for 30 min with shaking at 300 rpm. Protoplasts were harvested by centrifugation and lysed by resuspension in 200 µl PBS buffer. 15 µl of the sample was combined with 5 µl of 4× LDS sample buffer (NuPAGE), separated by SDS-PAGE, and transferred to PVDF membranes (Novex) using iBlot (Invitrogen). Blots were blocked in 5% skim milk (Carnation) in Tris-buffered saline (TBS)/Tween (TBS + 1% Tween 20; Sigma) overnight at 4°C with gentle shaking. Blots were incubated for 1 hr with antiserum raised against purified SpoIVA and detected using anti-rabbit IgG StarBright (Bio-Rad) with a ChemiDoc MP imager (BioRad).

## Mass determination (iSCAMS and SEC-MALS)

Mass Photometry (MP, iSCAMS) experiments were carried out on a OneMP instrument (Refeyn, Oxford, UK) at room temperature. Rectangular 24 × 50 mm coverslips (#12544E, Fisher Scientific) and square 24 × 24 mm coverslips (#1405–10, Globe Scientific) were prepared by rinsing with water, ethanol, and isopropanol, and dried with clean nitrogen gas (*Young et al., 2018*) Approximately 10 µl of protein was loaded into the channel formed by stacked coverslips. MP signals were recorded for 100 s to allow detection of at least $2 \times 10^3$ individual protein molecules. Raw MP data were processed in DiscoverMP software (Refeyn, Oxford, UK) and plotted as molar mass distribution histograms. For SEC-MALS, experiments were performed on a Agilent Series 1100 System (Agilent) with Superdex200 Increase 10/300 GL column (GE Healthcare), Helleos-II in-line multi angle light scattering detector (Wyatt Technology), and Optilab T-rEX refractive index detector (Wyatt Technology). SEC column was equilibrated with Buffer B until a stable refractive index baseline was reached. For sample analysis, 100 µl of SpoIVA at 0.94 mg/ml concentration in the presence of 4 mM ATP was injected at the 0.5 ml/min flow rate. All experiments were performed at room temperature, with MALS and RI detectors equilibrated at 20°C. Chromatograms were analyzed in ASTRA (V7.1, Wyatt Technology), and refractive index increment of 0.185 ml/g was used to determine the protein concentration.

## Electron microscopy

Negative staining of the protein samples was performed on glow-discharged carbon-coated grids. For each condition, 3.5 µl sample was applied to a grid and incubated for 40 s. Excess sample was blotted away using a filter paper. The grid was then stained with 3.5 µl 1% uranyl acetate solution for 1 min and air-dried for imaging. Digital micrographs were collected using a 2 k CCD camera on a Hitachi 7650 electron microscope at an accelerating voltage of 80kV.

## LC-MS quantification of metabolites

Cells were grown in CH media to $OD_{600nm}$ ~0.5 and induced to sporulate via the resuspension method as described above. Metabolite extraction was performed as described previously (*Yang et al., 2020*). Briefly, 10 ml culture were sampled and harvested by filtration through PTFE membrane (Sartorius) at time points before and after resuspension in A+B media. Pellets on the PTFE membranes were soaked in 3 ml extraction solvent mix (on ice 50:50 [v/v] chloroform/water) and then vortexed to quench metabolism and extract metabolites. Cell extracts were centrifuged at 5000 × g for 10 min to remove the organic phase, and then centrifuged at 20,000 × g for 10 min to remove cell debris. Samples were frozen at −80°C if not analyzed immediately. Samples were analyzed using LC-MS and the metabolites were quantified as described previously (*Fung et al., 2020*; *Yang et al., 2020*), using an HPLC-MS system consisting of a Vanquish UHPLC system linked to electrospray ionization (ESI, negative mode) to a Q Exactive Orbitrap mass spectrometer (Thermo Scientific) operated in full-scan mode to detect targeted metabolites based on their accurate masses. LC was performed on an Acquity UPLC BEH C18 column (1.7 µm, 2.1 × 100 mm; Waters). Total run time was 30 min with a flow rate of 0.2 ml/min, using Solvent A (97:3 [v/v] water/methanol, 10 mM tributylamine, and 10 mM acetic acid) and acetonitrile as Solvent B. The gradient was as follows: 0 min, 5% B; 2.5 min, 5% B; 19 min, 100% B; 23.5 min 100% B; 24 min, 5% B; 30 min, 5% B. Quantification of metabolites from raw LC-MS data was performed by using the MAVEN software (*Clasquin et al., 2012*). Normalized ion count was defined and calculated as the ion count per $OD_{600nm}$ per unit volume (5 ml) of the culture.

## Acknowledgements

We thank S Gottesman, S Wickner, M Maurizi, G Storz, A Khare, and D Court for discussions; Kunio Nagashima and Ziqiu Wang of the Electron Microscopy Laboratory of CCR for TEM sample preparation and imaging; and members of KSR lab for comments on the manuscript. This work was funded by the Intramural Research Program of the National Institutes of Health (NIH), National Cancer Institute, Center for Cancer Research (KSR), and National Library of Medicine (LA); NIH #R35GM127088 (JDW); and National Science Foundation #1715710 (DA-N).

## Additional information

### Funding

| Funder | Grant reference number | Author |
| --- | --- | --- |
| National Cancer Institute | Intramural Research Program | Kumaran Ramamurthi |
| National Institutes of Health | Intramural Research Program | L Aravind |
| National Institutes of Health | R35GM127088 | Jue D Wang |
| National Science Foundation | 1715710 | Daniel Amador-Noguez |

The funders had no role in study design, data collection and interpretation, or the decision to submit the work for publication.

### Author contributions

Taylor B Updegrove, Formal analysis, Investigation, Writing - original draft, Writing - review and editing; Jailynn Harke, Vivek Anantharaman, Jin Yang, Formal analysis, Investigation, Writing - review and editing; Nikhil Gopalan, Di Wu, David M Stevenson, Investigation; Grzegorz Piszczek, Formal analysis, Investigation; Daniel Amador-Noguez, Formal analysis, Funding acquisition; Jue D Wang, Formal analysis, Supervision, Funding acquisition, Writing - review and editing; L Aravind, Conceptualization, Formal analysis, Supervision, Writing - review and editing; Kumaran S Ramamurthi, Conceptualization, Formal analysis, Supervision, Funding acquisition, Methodology, Writing - original draft, Project administration, Writing - review and editing

### Author ORCIDs

Vivek Anantharaman http://orcid.org/0000-0001-8395-0009
Jue D Wang http://orcid.org/0000-0003-1503-170X
L Aravind http://orcid.org/0000-0003-0771-253X
Kumaran S Ramamurthi https://orcid.org/0000-0002-2335-3568

### Decision letter and Author response

Decision letter https://doi.org/10.7554/eLife.65845.sa1
Author response https://doi.org/10.7554/eLife.65845.sa2

## Additional files

### Supplementary files

• Source data 1. Summary of calculated data for in vitro assays. For *Figures 2*, *4,* and *5*.

• Supplementary file 1. *Bacillus subtilis* strains used in this study.

• Supplementary file 2. Summary of results from in vitro and in vivo assays performed with various SpoIVA variants. For subcellular localization data, '+' indicates forespore localization pattern qualitatively similar to wild-type GFP-SpoIVA; '-' indicates mis-localization. Errors are S.D.

• Supplementary file 3. Nucleotide levels (ion counts) at indicated time points after induction of sporulation in *B. subtilis* via resuspension method. 0 hr time point indicates time of sporulation induction by resuspension; pre-induction is immediately prior to resuspension. Errors are S.D.

• Transparent reporting form

### Data availability

All data generated or analysed during this study are included in the manuscript and supporting files. Source data files have been provided for Figures 2 and 6.

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
