## [Decision Letter]

**Acceptance summary:**

In this study, Updegrove et al., determined how and why the key sporulation protein SpoIVA from *Bacillus subtilis* evolved from an ancestral enzyme with GTPase activity towards the ATPase it is nowadays. The authors identified changes in active site amino acids that accomplished the change in nucleotide specificity. An engineered SpoIVA variant functioned as GTPase but failed to polymerize, suggesting an important role of the nucleotide base in this biological function. The authors propose that increased ATP relative to GTP levels at the end of sporulation drove the evolutionary conversion of SpoIVA towards its novel nucleotide preference.

**Decision letter after peer review:**

[Editors’ note: the authors submitted for reconsideration following the decision after peer review. What follows is the decision letter after the first round of review.]

Thank you for submitting your work entitled "Resurrection of ancestral GTPase activity in an extant ATPase reveals a nucleotide base requirement for function" for consideration by *eLife*. Your article has been reviewed by a Senior Editor, a Reviewing Editor, and three reviewers. The following individual involved in review of your submission has agreed to reveal their identity: Peter Setlow (Reviewer #2).

Our decision has been reached after consultation between the reviewers. Based on these discussions and the individual reviews below, we regret to inform you that your work will not be considered further for publication in *eLife*.

Indeed, while all of us thought that your study addresses an interesting research question and that the mutagenesis and general characterization of the mutant proteins was well performed, the expert reviewers identified several shortcomings in your study. These shortcomings were on one hand related to missing experimental data that seem essential to support the conclusions of your manuscript and on the other hand the somewhat biased experimental setup.

While ancestor resurrection can be a powerful approach to provide insights into protein evolution and specialization, your study was not considered to follow the usual approaches that allow the identification of ancestral proteins. Indeed, your exclusive focus on the active site of SpoIVA could have biased your study, as such a targeted analysis ignores the periphery of the nucleotide binding site and the co-evolving protein as a whole (e.g., more commonly used resurrection approaches are based on phylogeny and statistical methods).

In addition, the lack of mechanistic insight regarding the biochemical differences between SpoIVA and SpoIVA-resurrected as well as lack of mechanistic insight of the oligomerization step makes it difficult to conclude that the different nucleotide bases cause this effect compared to other possibilities (e.g., lower ATPase activity, other changes introduced by the mutations, whether similar ATPase-dependent intermediates are observable or not, etc).

The experts also missed experiments with ADP/GDP, non-hydrolyzing analogues of ATP and GTP (e.g. ATPgammaS or GTPgammaS), and potentially also transition state analogues, which seems to be essential to distinguish between conformational changes/protection from proteolysis versus merely nucleotide binding.

Lastly, it was noted that a primary argument used in your manuscript for the change in NTP preference (from GTP to ATP) is based on a drop in the GTP levels after the onset of sporulation. However, the published data on this aspect seem to have addressed a time point of max. 2 hr after sporulation induction, which might be before SpoIVA's function. Getting definitive data for levels of ATP/GTP in the sporulating cell at around the time that SpoIVA is active would therefore be crucial to support your central argument. While the reviewers appreciate that these experiments are not trivial, definitive data on this point using current technology could be important not only for this manuscript but for the sporulation field as a whole.

Reviewer #1:

In this study, Updegrove and colleagues investigated the role of ATP versus GTP hydrolysis in the oligomerization of SpoIVA during *B. subtilis* sporulation. Authors form the same group previously showed that SpoIVA contains a TRAFAC class GTPase domain that uses ATP hydrolysis rather than GTP hydrolysis to perform its function. They further previously showed that ATP hydrolysis rather than ATP binding is required to induce conformational changes in the SpoIVA subunit that drive incorporation into nucleotide-free SpoIVA polymeric filaments. In the current study the authors re-engineered the binding pocket of SpoIVA toward the (presumed) ancestral counterpart displaying equal catalytic efficiency (k_cat_/K_M_) toward ATP and GTP. Interestingly they find that, unlike ATP hydrolysis, GTP hydrolysis does not support SpoIVA polymerization in vitro. This indicates that apparently also the nucleotide base plays a role in protein function, rather than just the energy release from phosphodiester bond hydrolysis. Although this is a very interesting finding per se, mechanistic insights regarding either the processes leading to protein oligomerization or the presumed role of the nucleotide base in this process are rather limited. I also have a number of questions and doubts regarding the experimental support for some of the claims that are being made, as outlined below.

1) It is not entirely clear to me what the added value is of the re-engineering toward an ancestral protein with equal k_cat_/K_M_ values for GTP and ATP for the final conclusions. Wild-type SpoIVA displays only a relatively marginal preference (somewhat more than three-fold) for ATP over GTP? Moreover, wild-type SpoIVA in fact uses GTP more efficiently than the resurrected SpoIVA. Why didn't the authors just use wild-type SpoIVA to show that GTP does not induce polymerization in vitro?

2) In Figure 2—figure supplement 1 the Michaelis-Menten curves for GTP and ATP hydrolysis are shown. Also, the resulting k_cat_ and K_M_ values should be reported (rather than just k_cat_/K_M_).

From the curves it seems that the K_M_ values for ATP and GTP hydrolysis are rather severely affected for the resurrected SpoIVA (NSxD SxA), potentially leading to a large error on the fitted parameters (due to saturation not being reached). Under these circumstances one can also doubt how much of the protein is being bound under the substrate conditions used in most other experiments (generally 4 mM ATP or GTP is used which seems to be not saturating for ATP).

3) Subsection “ATP, but not GTP, hydrolysis drives in vitro polymerization of resurrected SpoIVA*”* and Figure 4A/B. The authors make the observation that mutants NKxR SxE, NKxD SxE, NKxD AxA are not able to utilize the energy released from hydrolysis of either ATP or GTP to drive conformational changes in the protein. This while these mutants hydrolyze ATP or GTP equally well, and even better, than wild-type SpoIVA. This observation is both rather strange and intriguing. However, to prove that the conformational change and protection from proteolysis is really due to hydrolysis and not merely to nucleotide binding, it is required that this experiment is also performed in presence of ADP/GDP, non-hydrolyzing analogues of ATP and GTP (e.g. ATPgammaS or GTPgammaS), and potentially also transition state analogues such as ADP.AlFx and GDP.AlFx.

Related to the remark above: the experiments shown in Figure 4A are done with 4 mM ATP or GTP and thus under multiple turnover conditions. This means that every enzyme molecule presumably went through subsequent rounds of substrate turnover. The main nucleotide state the protein resides in at any particular moment would thus depend on the relative rate of substrate binding, chemical turnover and product release. Could the authors comment on the main nucleotide state SpoIVA would be in during limited proteolysis?

4) Subsection “Formation of an ADP-bound SpoIVA multimeric intermediate is required for polymerization”. The authors report that addition of ATP to SpoIVA, below its critical concentration of polymerization, resulted to a shift of the peak on SEC toward the void volume.

– This seems to contradict what was reported in Castaing et al., 2013, where the authors (from the same group) report that "presence or absence of ATP did not affect the oligomerization state of IVA" at a concentration of 2µM.

– The SEC-MALS experiment should also be performed in presence of ADP (and preferentially a non-hydrolysable ATP analogue, which differs from the strategy of using a protein variant where one of the main crucial switch residues has been mutated!).

– Subsection “Formation of an ADP-bound SpoIVA multimeric intermediate is required for polymerization” and Figure 5—figure supplement 1C: In the SEC-MALS experiment the authors find a peak (see above) in the void volume, and the MALS analysis reveals a varying range of molecular masses ranging from 10E6 to 5.10E3 kDa. The authors interpret this as "at least 36 monomers" and in Figure 7 this becomes ∼ 16 dimers. Using a subunit molecular mass of 57kDA, a quick calculation shows that the peak would contain species ranging between 35 and about 15000 subunit copies, which might just as well correspond to aggregates rather than an "on-pathway" intermediate. The authors could for example use (negative stain) EM to investigate the nature of the species in the void volume and make that distinction.

Reviewer #2:

This paper describes work on the SpoIVA protein essential for spore coat assembly during sporulation of the bacterial spore-former *Bacillus subtilis*. SpoIVA uses ATP hydrolysis to drive its polymerization on the outside of the developing spore, providing a framework for coat assembly. However, SpoIVA appears to have evolved from an ancestor that preferentially used GTP not ATP. The authors have made multiple amino acid changes in residues involved in NTP hydrolysis/recognition, converting present SpoIVA to a protein using GTP as well as ATP – at least in vitro. The catalytic efficiency of the wt and ultimate mutant (SpoIVAresurrected) protein in vitro, was ~4-fold higher with ATP than GTP for wt SpoIVA but ~1 for SpoIVAresurrected. This difference is likely bigger in vivo since intracellular [ATP] is ~ 5-fold higher than [GTP]. Since [GTP] is reported to fall significantly in sporulation, the authors suggest that this factor is what drove the evolution of SpoIVA to preferentially use ATP.

The authors' mutagenesis, analyses of wt and mutant proteins' ATP and GTP hydrolysis in vitro at different [NTP], in vitro polymerization of proteins with either NTP, and sporulation efficiency with the wt, resurrected strain and other mutants all seems well done. Surprisingly, the resurrected SpoIVA protein completely complemented sporulation of a spoIVA mutant.

1) Following all the data in the ms and drawing conclusions was difficult since there was no Table summarizing mutants' behavior in vivo and in vitro. It was also difficult to assess the likely preference of ATP/GTP in vivo, as presumably the catalytic efficiencies reported for the variants in vitro is at saturating [ATP] and [GTP] and in vivo [GTP] is usually ~5-fold lower than [ATP] and presumably much less early in sporulation (see #2). Overall, it was not easy to see why SpoIVA likely preferred ATP over GTP in sporulation for both the wt and resurrected strain. A Table summarizing various properties of various SpoIVA proteins in vitro and in vivo would be helpful to the reader.

2) The basis for suggesting that low GTP levels in sporulation drove changes in NTP preference in SpoIVA is work from Ernst Freese years ago that showed significant drops in [GTP] upon sporulation induction by various conditions, and this may be correct. However, this work reported results only for 2 hr after sporulation induction, and probably before SpoIVA functions. Whether GTP levels are low this late in sporulation is not known. Indeed, since protein and RNA synthesis continue in the mother cell during sporulation, GTP must be continuously available. Unfortunately, there are no analyses of NTPs' levels throughout sporulation.

3) The NTP levels determined by Freese were from culture samples filtered and then the filters floated on formic acid; this took ~8 sec. While values for energy charge (ATP + 0.5 ADP/ATP + AMP + ADP) were what one would have expected for growing bacteria, if GTP turns over more rapidly than ATP, GTP levels determined might be lower than they really are.

Reviewer #3:

In the manuscript, Updegrove et al., use the unique ATPase SpoIVA from *Bacillus subtilis* as a model to investigate a fundamental question: Does nucleotide identity dictate hydrolysis-driven protein function? In previous work, the authors pointed out that TRAFAC GTPases, and in particular bacterial Era, likely share a common ancestor. SpoIVA is a cytoskeletal protein that uses its ATPase activity to form terminal polymers in an initial step of Bacillus spore formation. Hence, it appears that SpoIVA evolved ATPase activity from a GTPase scaffold to cope with the cellular conditions met during spore formation. Specifically, as the authors argue in the current manuscript, this change may have been driven by limiting GTP concentrations under conditions that drive spore formation, whereas ATP would still be available at higher levels under the same conditions.

To investigate the mechanistic consequences of turning SpoIVA into an Era-like GTPase, targeted mutagenesis at well-established signature motifs that determine substrate preference in these enzymes was employed. One particular mutant, called SpoIVA-resurrected, does not discriminate between ATP or GTP, hydrolyzing both substrates at near-equal efficiency and at efficiencies similar to Era (although with reduced ATPase activity compared to wild-type SpoIVA). In cells, SpoIVA-resurrected supports sporulation at levels similar to wild-type SpoIVA. In tryptic digests, wild-type and "resurrected" SpoIVA are both stabilized by ATP and GTP. For wild-type SpoIVA, polymerization (above a critical protein concentration for polymerization) and an intermediate (below a critical protein concentration) was detected only with ATP. The intermediate was shown to corresponds to an ADP-bound, post-hydrolysis state. Finally, it was shown that SpoIVA-resurrected has comparatively reduced sensitivity to ATP concentrations, with wild-type SpoIVA polymerizing at much lower ATP levels.

While the work addresses a fundamental question, major concerns regarding the approach used to investigate the evolution of functional specialization of GTPases and ATPases dampen enthusiasm for this study. Also, the molecular mechanism responsible for the biochemical differences between SpoIVA and SpoIVA-resurrected remains enigmatic, which, together with alternative explanations for the apparent differences in polymerzation and other caveats, may limit the perceived impact.

1) Ancestor resurrection has become a popular method for rationalizing the evolution of structural and/or functional properties in related proteins. Usually, the approach involves statistical methods to predict the evolution of proteins or domains based on phylogeny, taking into account conservation across their entire sequence. Here, the authors focus exclusively on functional motifs in the active site of SpoIVA (and Era). This is a fairly targeted analysis that does not take into account residues in the periphery of the nucleotide binding site (or the protein as a whole) that could contribute to substrate specificity, catalytic efficiency, and/or switching. One concern is that by doing so, the authors may miss residues or regions in the protein, which may co-evolve with the active-site motifs, shaping the functional properties of the entire protein.

It is recommended to use one of the now common approaches to resurrect the last common ancestor of SpoIVA and Era (see e.g. 10.1146/annurev-biophys-070816-033631), and study its properties in an unbiased fashion (e.g., What sequence motifs emerge? What is its activity and substrate profile? Does it polymerize and/or form hydrolysis-dependent oligomers?).

2) The authors use Era, "ancestral Era GTPase", and the ancestor interchangeably (e.g., subsection “The altered NKxD motif in G4 and SxE sequence in G5 mediate nucleotide specificity of SpoIVA”, but common throughout the manuscript). However, Era appears to be an extant enzyme that likely evolved independently after the emergence of SpoIVA, which may have further contributed to the specialization of the two proteins compared to a common ancestor. Careful separation between extant Era and SpoIVA and predicted common ancestors would avoid the potential for confusion.

3) If the targeted changes in the primary active site motifs fully recapitulate the evolutionary and functional differences between SpoIVA and Era, one may predict that the corresponding changes in Era would result in SpoIVA-like properties. This would be a crucial experiment to consider for this study.

4) While the type of mutations introduced into SpoIVA was guided by sequence comparison with Era-type GTPases, it appears that the authors chose the mutant they refer to as the resurrected ancestor based on the activity profile that matches most closely the catalytic characteristics of Era, and not necessarily based on the phylogenic analysis. This practice introduces bias that may hamper the analysis concerning the evolution of substrate specificity.

In this context, it is also not clear why the authors assume that the last common ancestor of SpoIVA and Era had Era-like hydrolysis efficiencies and substrate preference.

5) The authors also did not consider the potential impact the register shift in G5 between SpoIVA (sequence SxE) and Era (sequence SA) may have, only focusing on the sequence register found in SpoIVA. It is conceivable that the additional residue in this motif arose to accommodate an ATP preference and ATPase-driven mechanism.

6) It is not clear whether the defects in polymerization characteristics of the SpoIVA-resurrected mutant can be attributed to the lower ATPase activity compared to wild-type SpoIVA (or other subtle changes introduced by the mutations). It is also not clear whether SpoIVA-resurrected forms a similar ATPase-dependent intermediate that was oberserved with wild-type SpoIVA (at protein concentrations below the threshold for polymerization). This is an important experiment since the differences between ATP and GTP in their capacity to drive intermediate formation could be explained by the different ATPase vs GTPase efficiencies of the wild-type protein; SpoIVA-resurrected has overall lower catalytic activity compared to wild-type, and it does not support polymerization. Hence, there is a possibility that the observed differences can be attributed to catalytic efficiency (maybe in addition to the nature of the nucleotide base).

---

## [Author Response]

[Editors’ note: the authors resubmitted a revised version of the paper for consideration. What follows is the authors’ response to the first round of review.]

Indeed, while all of us thought that your study addresses an interesting research question and that the mutagenesis and general characterization of the mutant proteins was well performed, the expert reviewers identified several shortcomings in your study. These shortcomings were on one hand related to missing experimental data that seem essential to support the conclusions of your manuscript and on the other hand the somewhat biased experimental setup.

We thank the reviewers, who clearly represent a broad spectrum of expertise, for their time rigorously reviewing our work and offering valuable suggestions that we believe have enhanced the manuscript. We have incorporated every suggestion that they proposed, and we think that the result is a greatly improved manuscript that we hope will be of interest to the broad readership of *eLife*. Included below are our point-by-point responses to each of the reviewers’ suggestions.

While ancestor resurrection can be a powerful approach to provide insights into protein evolution and specialization, your study was not considered to follow the usual approaches that allow the identification of ancestral proteins. Indeed, your exclusive focus on the active site of SpoIVA could have biased your study, as such a targeted analysis ignores the periphery of the nucleotide binding site and the co-evolving protein as a whole (e.g., more commonly used resurrection approaches are based on phylogeny and statistical methods).

We understand reviewer 3’s motivation to follow the “usual” approaches to ancestor reconstruction, but such an approach is not technically feasible for the kind of evolutionary divergence involved for SpoIVA and Era. We have provided a more detailed response to this set of concerns below and have now more clearly explained our justification for the approach that we chose in the Results section. Additionally, we have made two changes that clarify our approach:

1) We have modified Figure 1C to accentuate the fact that SpoIVA contains additional (unique) domains compared to Era (the gray ovals for the additional domains have been enlarged).

2) We understand that the term “resurrection” might have been construed as following a specific computational approach. Hence, in the revised manuscript we have completely eschewed the use of the word and have instead used “reformulation” (including in the title) or “reengineered” to more accurately represent directed changes that we introduced to the active site of the enzyme.

In addition, the lack of mechanistic insight regarding the biochemical differences between SpoIVA and SpoIVA-resurrected as well as lack of mechanistic insight of the oligomerization step makes it difficult to conclude that the different nucleotide bases cause this effect compared to other possibilities (e.g., lower ATPase activity, other changes introduced by the mutations, whether similar ATPase-dependent intermediates are observable or not, etc).The experts also missed experiments with ADP/GDP, non-hydrolyzing analogues of ATP and GTP (e.g. ATPgammaS or GTPgammaS), and potentially also transition state analogues, which seems to be essential to distinguish between conformational changes/protection from proteolysis versus merely nucleotide binding.

In the revised version, we have performed all the experiments that reviewer 1 suggested, including examining the gross ultrastructure of the assembly intermediate using TEM, demonstrating that the reformulated version of SpoIVA forms the assembly intermediate. We also extended our initial observations using ATPase mutants, now with wild type protein using non-hydrolyzable nucleotide analogs and transition state analogs, as reviewer 1 suggested. We thank the reviewer for the transition state analog experiments which were invaluable for constructing a more detailed mechanistic model for SpoIVA polymerization that provides the most detailed description to date of how SpoIVA polymerizes.

Lastly, it was noted that a primary argument used in your manuscript for the change in NTP preference (from GTP to ATP) is based on a drop in the GTP levels after the onset of sporulation. However, the published data on this aspect seem to have addressed a time point of max. 2 hr after sporulation induction, which might be before SpoIVA's function. Getting definitive data for levels of ATP/GTP in the sporulating cell at around the time that SpoIVA is active would therefore be crucial to support your central argument. While the reviewers appreciate that these experiments are not trivial, definitive data on this point using current technology could be important not only for this manuscript but for the sporulation field as a whole.

As the Editor suggested, these experiments were indeed challenging, but ultimately worth the added effort. In the revised manuscript, in collaboration with Profs. Jade Wang and Daniel Amador-Noguez (University of Wisconsin), we report the levels and calculated intracellular concentrations of ATP, GTP, CTP, UTP, ADP, GDP, CDP, UDP, AMP, GMP, CMP, UMP, and the nucleotide alarmones ppGpp and pGpp immediately prior to and after induction of sporulation, and at t = 1, 2, 3.5, and 5 h after sporulation induction. The results, presented in a new Figure 6 revealed that ATP levels are indeed ~5-8-fold higher than GTP, depending on the time of sporulation. Moreover, we show that the reduced level of ATP promoted robust polymerization of the extant (WT) SpoIVA, but not the reengineered SpoIVA^GTPase^, suggesting that, along with scarcity of GTP, reduced ATP levels could provide the selective pressure that drove SpoIVA to optimally utilize ATP. We agree that this data will be useful to the sporulation community as a whole.

Reviewer #1:In this study, Updegrove and colleagues investigated the role of ATP versus GTP hydrolysis in the oligomerization of SpoIVA during *B. subtilis* sporulation. Authors form the same group previously showed that SpoIVA contains a TRAFAC class GTPase domain that uses ATP hydrolysis rather than GTP hydrolysis to perform its function. They further previously showed that ATP hydrolysis rather than ATP binding is required to induce conformational changes in the SpoIVA subunit that drive incorporation into nucleotide-free SpoIVA polymeric filaments. In the current study the authors re-engineered the binding pocket of SpoIVA toward the (presumed) ancestral counterpart displaying equal catalytic efficiency (k_cat_/K_M_) toward ATP and GTP. Interestingly they find that, unlike ATP hydrolysis, GTP hydrolysis does not support SpoIVA polymerization in vitro. This indicates that apparently also the nucleotide base plays a role in protein function, rather than just the energy release from phosphodiester bond hydrolysis. Although this is a very interesting finding per se, mechanistic insights regarding either the processes leading to protein oligomerization or the presumed role of the nucleotide base in this process are rather limited. I also have a number of questions and doubts regarding the experimental support for some of the claims that are being made, as outlined below.

We are pleased that the reviewer thought our findings were interesting. In the revised version of the manuscript, we have incorporated every suggestion from the reviewer to provide more mechanistic insight into the trajectory of nucleotide binding and hydrolysis and how each event contributes to SpoIVA polymerization.

1) It is not entirely clear to me what the added value is of the re-engineering toward an ancestral protein with equal k_cat_/K_M_ values for GTP and ATP for the final conclusions. Wild-type SpoIVA displays only a relatively marginal preference (somewhat more than three-fold) for ATP over GTP? Moreover, wild-type SpoIVA in fact uses GTP more efficiently than the resurrected SpoIVA. Why didn't the authors just use wild-type SpoIVA to show that GTP does not induce polymerization in vitro?

We agree this data should be highlighted a bit more in our text. After establishing this observation, as the reviewer suggested, we attempted to build a more mechanistic explanation as to why SpoIVA fails to polymerize with GTP by utilizing the reformulation variants.

To divulge more of our logic for clarity, we restored the ancestral enzymatic site with the prediction that energy expended by hydrolysis of any nucleotide would permit SpoIVA to polymerize since reengineering nucleotide specificity in other motor proteins has been shown previously to retain their motor function regardless of which nucleotide is used to release the energy. Hence, we were initially surprised that SpoIVA^GTPase^ did not polymerize. The rest of work sought to explain this curious finding and provide mechanistic insights by utilizing the reformulation variants.

2) In Figure 2—figure supplement 1 the Michaelis-Menten curves for GTP and ATP hydrolysis are shown. Also, the resulting k_cat_ and K_M_ values should be reported (rather than just k_cat_/K_M_).

We agree and have now reported the calculated *k*_cat_ and *K*_m_ values for each variant in a new Supplementary file 2.

From the curves it seems that the K_M_ values for ATP and GTP hydrolysis are rather severely affected for the resurrected SpoIVA (NSxD SxA), potentially leading to a large error on the fitted parameters (due to saturation not being reached). Under these circumstances one can also doubt how much of the protein is being bound under the substrate conditions used in most other experiments (generally 4 mM ATP or GTP is used which seems to be not saturating for ATP).

We have re-purified this variant and re-performed this analysis multiple times, and the results are represented in a new Figure 2—figure supplement 1-I. Our displayed plot in Figure 2—figure supplement 1-I now more clearly displays Michaelis-Menten kinetics. With multiple additional replicates, the NSxD SxA SpoIVA^GTPase^ variant also shows a slight preference for GTP hydrolysis.

3) Subsection “ATP, but not GTP, hydrolysis drives in vitro polymerization of resurrected SpoIVA” and Figure 4A/B. The authors make the observation that mutants NKxR SxE, NKxD SxE, NKxD AxA are not able to utilize the energy released from hydrolysis of either ATP or GTP to drive conformational changes in the protein. This while these mutants hydrolyze ATP or GTP equally well, and even better, than wild-type SpoIVA. This observation is both rather strange and intriguing.

We agree and were also initially baffled by this result.

However, to prove that the conformational change and protection from proteolysis is really due to hydrolysis and not merely to nucleotide binding, it is required that this experiment is also performed in presence of ADP/GDP, non-hydrolyzing analogues of ATP and GTP (e.g. ATPgammaS or GTPgammaS), and potentially also transition state analogues such as ADP.AlFx and GDP.AlFx.

We have performed the requested experiments and the results are now displayed in a new Figure 4C-F, where we compare conformational changes in SpoIVA after exposure to ATP/GTP, ATP-γ-S/GTP-γ-S, ADP/GDP, and ADP-AlFx/GDP-AlFx; the data are now discussed extensively in a new paragraph (subsection “Hydrolysis of either ATP or GTP can drive a conformational change in SpoIVA”). The results of the experiment now permit the conclusion that simply binding to ADP or GDP is insufficient to produce the full conformational change in SpoIVA required for polymerization. Instead, SpoIVA polymerization requires two things: (1) the hydrolysis of an NTP molecule, and (2) retention of the substrate within the active site, as evidenced by the ability of ADP-AlFx/GDP-AlFx, but not ADP/GDP, to produce a conformational change similar to ATP/GTP hydrolysis. Additionally, the data allowed us to suggest that rapid turnover (hydrolysis and release) of an NTP is incompatible with producing the conformational change. This turned out to be an incredibly informative set of experiments- we thank the referee for suggesting it.

Related to the remark above: the experiments shown in Figure 4A are done with 4 mM ATP or GTP and thus under multiple turnover conditions. This means that every enzyme molecule presumably went through subsequent rounds of substrate turnover. The main nucleotide state the protein resides in at any particular moment would thus depend on the relative rate of substrate binding, chemical turnover and product release. Could the authors comment on the main nucleotide state SpoIVA would be in during limited proteolysis?

Since the proteolysis experiment occurs after 4 hours of exposure to excess ATP or GTP, and at a concentration below the critical concentration for polymerization, the SpoIVA in these experiments is likely in the ADP/GDP bound state.

4) Subsection “Formation of an ADP-bound SpoIVA multimeric intermediate is required for polymerization” The authors report that addition of ATP to SpoIVA, below its critical concentration of polymerization, resulted to a shift of the peak on SEC toward the void volume.– This seems to contradict what was reported in Castaing et al., 2013, where the authors (from the same group) report that "presence or absence of ATP did not affect the oligomerization state of IVA" at a concentration of 2µM.

We also recognized this discrepancy, which is what prompted us to modify the polymerization model to include an ADP-bound, higher molecular weight, polymerization-competent intermediate. We can provide three possible explanations to explain the difference. First, compared to Castaing et al., the SpoIVA purification protocol is slightly different. After Ni^2+^-affinity chromatography, ion exchange chromatography yields three separate peaks of SpoIVA. In 2013 (Castaing et al.,) we determined that the SpoIVA population in the third peak polymerized most robustly in an ATP-dependent manner and therefore used that fraction for the report. Later, we discovered, using DLS and negative stain TEM, that peak 3 was also the most heterogeneous in terms of size and hydrodynamic radius, suggesting that this fraction may have already formed the polymerization intermediate (which may be why it readily polymerized). The second peak yielded largely homogeneous SpoIVA that was clearly a dimer, which is the fraction that we have used in this study, and which clearly formed the heterogeneous polymerization-competent assembly intermediate upon ATP addition. Secondly, in Castaing et al., we used a Superdex 200 size exclusion column (MW range: 10,000 to 600,000 Da) to separate SpoIVA products after the size exclusion chromatography step; in the current report we used a Superose 6 column (MW range: 5,000-5,000,000 Da), which has a larger separation range. A final possible explanation, which is likely less relevant, is that the Mg^2+^ concentration used for polymerization reactions differed. In Castaing et al., (and earlier in Ramamurthi et al., 2008), we used 10 mM MgCl2 for polymerization reactions. At this concentration, polymerization reactions were routinely inconsistent in that some preparations of the wild type protein would not polymerize. At that time, we (incorrectly) attributed this to batch-to-batch variations in protein purification. Subsequent to those reports, we carefully titrated each buffer component and discovered that 5 mM MgCl2 in the reaction buffer surprisingly gave us extremely consistent polymerization reactions. Thus, a combination of improved protein purification techniques and reaction conditions allowed us in this to more carefully isolate reaction intermediates.

– The SEC-MALS experiment should also be performed in presence of ADP (and preferentially a non-hydrolysable ATP analogue, which differs from the strategy of using a protein variant where one of the main crucial switch residues has been mutated!).

We agree and we thank the reviewer for suggesting this experiment. In the revised Figure 5B-C, we now report that incubation of WT SpoIVA with either ADP or the non-hydrolyzable ATP-γ-S analogue of ATP abrogates formation of the ADP-bound polymerization intermediate. Taken together with the inability of the SpoIVA sensor T-variant that is able to bind, but not hydrolyze, ATP, we can now more confidently conclude that ATP hydrolysis is a requirement for formation of the polymerization intermediate.

– Subsection “Formation of an ADP-bound SpoIVA multimeric intermediate is required for polymerization” and Figure 5—figure supplement 1C: In the SEC-MALS experiment the authors find a peak (see above) in the void volume, and the MALS analysis reveals a varying range of molecular masses ranging from 10E6 to 5.10E3 kDa. The authors interpret this as "at least 36 monomers" and in Figure 7 this becomes ∼ 16 dimers. Using a subunit molecular mass of 57kDA, a quick calculation shows that the peak would contain species ranging between 35 and about 15000 subunit copies, which might just as well correspond to aggregates rather than an "on-pathway" intermediate. The authors could for example use (negative stain) EM to investigate the nature of the species in the void volume and make that distinction.

We have now analyzed the void volume fraction by TEM. The images, reported in a new Figure 5D, indicate somewhat regularly shaped particles that clump together in solution, which may explain the variability of size in this fraction.

Reviewer #2:This paper describes work on the SpoIVA protein essential for spore coat assembly during sporulation of the bacterial spore-former *Bacillus subtilis*. SpoIVA uses ATP hydrolysis to drive its polymerization on the outside of the developing spore, providing a framework for coat assembly. However, SpoIVA appears to have evolved from an ancestor that preferentially used GTP not ATP. The authors have made multiple amino acid changes in residues involved in NTP hydrolysis/recognition, converting present SpoIVA to a protein using GTP as well as ATP – at least in vitro. The catalytic efficiency of the wt and ultimate mutant (SpoIVAresurrected) protein in vitro, was ~4-fold higher with ATP than GTP for wt SpoIVA but ~1 for SpoIVAresurrected. This difference is likely bigger in vivo since intracellular [ATP] is ~ 5-fold higher than [GTP]. Since [GTP] is reported to fall significantly in sporulation, the authors suggest that this factor is what drove the evolution of SpoIVA to preferentially use ATP.The authors' mutagenesis, analyses of wt and mutant proteins' ATP and GTP hydrolysis in vitro at different [NTP], in vitro polymerization of proteins with either NTP, and sporulation efficiency with the wt, resurrected strain and other mutants all seems well done. Surprisingly, the resurrected SpoIVA protein completely complemented sporulation of a spoIVA mutant.1) Following all the data in the ms and drawing conclusions was difficult since there was no Table summarizing mutants' behavior in vivo and in vitro. It was also difficult to assess the likely preference of ATP/GTP in vivo, as presumably the catalytic efficiencies reported for the variants in vitro is at saturating [ATP] and [GTP] and in vivo [GTP] is usually ~5-fold lower than [ATP] and presumably much less early in sporulation (see #2). Overall, it was not easy to see why SpoIVA likely preferred ATP over GTP in sporulation for both the wt and resurrected strain. A Table summarizing various properties of various SpoIVA proteins in vitro and in vivo would be helpful to the reader.

We agree and have now included a new Table S2 that summarizes all of the in vivo and in vitro data for the behavior of the different SpoIVA variants. In addition, as requested by reviewer 1, Supplementary file 2 also contains the *K*_m_ and *K*_cat_ values used to calculate the catalytic efficiencies.

2) The basis for suggesting that low GTP levels in sporulation drove changes in NTP preference in SpoIVA is work from Ernst Freese years ago that showed significant drops in [GTP] upon sporulation induction by various conditions, and this may be correct. However, this work reported results only for 2 hr after sporulation induction, and probably before SpoIVA functions. Whether GTP levels are low this late in sporulation is not known. Indeed, since protein and RNA synthesis continue in the mother cell during sporulation, GTP must be continuously available. Unfortunately, there are no analyses of NTPs' levels throughout sporulation.

In the revised manuscript, we utilized LC-MS to measure ATP and GTP levels at various time points after induction of sporulation. In addition, we assessed the levels of CTP, UTP, ADP, GDP, CDP, UDP, AMP, GMP, CMP, UMP, and the nucleotide alarmones ppGpp and pGpp. These data are presented in a new Figure 6C-6F and show that ATP levels are ~8-fold higher than that of GTP at t=3.5 hours after induction of sporulation, when SpoIVA is maximally active. ATP levels remain ~5-fold higher at t=5 hours. The data also provide a global landscape of nucleotide availability as sporangia proceed to dormancy that we hope will be a valuable reference for sporulation researchers.

3) The NTP levels determined by Freese were from culture samples filtered and then the filters floated on formic acid; this took ~8 sec. While values for energy charge (ATP + 0.5 ADP/ATP + AMP + ADP) were what one would have expected for growing bacteria, if GTP turns over more rapidly than ATP, GTP levels determined might be lower than they really are.Reviewer #3:In the manuscript, Updegrove et al., use the unique ATPase SpoIVA from *Bacillus subtilis* as a model to investigate a fundamental question: Does nucleotide identity dictate hydrolysis-driven protein function? In previous work, the authors pointed out that TRAFAC GTPases, and in particular bacterial Era, likely share a common ancestor. SpoIVA is a cytoskeletal protein that uses its ATPase activity to form terminal polymers in an initial step of Bacillus spore formation. Hence, it appears that SpoIVA evolved ATPase activity from a GTPase scaffold to cope with the cellular conditions met during spore formation. Specifically, as the authors argue in the current manuscript, this change may have been driven by limiting GTP concentrations under conditions that drive spore formation, whereas ATP would still be available at higher levels under the same conditions.To investigate the mechanistic consequences of turning SpoIVA into an Era-like GTPase, targeted mutagenesis at well-established signature motifs that determine substrate preference in these enzymes was employed. One particular mutant, called SpoIVA-resurrected, does not discriminate between ATP or GTP, hydrolyzing both substrates at near-equal efficiency and at efficiencies similar to Era (although with reduced ATPase activity compared to wild-type SpoIVA). In cells, SpoIVA-resurrected supports sporulation at levels similar to wild-type SpoIVA. In tryptic digests, wild-type and "resurrected" SpoIVA are both stabilized by ATP and GTP. For wild-type SpoIVA, polymerization (above a critical protein concentration for polymerization) and an intermediate (below a critical protein concentration) was detected only with ATP. The intermediate was shown to corresponds to an ADP-bound, post-hydrolysis state. Finally, it was shown that SpoIVA-resurrected has comparatively reduced sensitivity to ATP concentrations, with wild-type SpoIVA polymerizing at much lower ATP levels.While the work addresses a fundamental question, major concerns regarding the approach used to investigate the evolution of functional specialization of GTPases and ATPases dampen enthusiasm for this study. Also, the molecular mechanism responsible for the biochemical differences between SpoIVA and SpoIVA-resurrected remains enigmatic, which, together with alternative explanations for the apparent differences in polymerzation and other caveats, may limit the perceived impact.

We thank the reviewer for appreciating the fundamental question we are trying to address and have addressed each of the specific concerns that they raised with additional experiments to more clearly reveal a molecular mechanism of SpoIVA polymerization

1) Ancestor resurrection has become a popular method for rationalizing the evolution of structural and/or functional properties in related proteins. Usually, the approach involves statistical methods to predict the evolution of proteins or domains based on phylogeny, taking into account conservation across their entire sequence. Here, the authors focus exclusively on functional motifs in the active site of SpoIVA (and Era). This is a fairly targeted analysis that does not take into account residues in the periphery of the nucleotide binding site (or the protein as a whole) that could contribute to substrate specificity, catalytic efficiency, and/or switching. One concern is that by doing so, the authors may miss residues or regions in the protein, which may co-evolve with the active-site motifs, shaping the functional properties of the entire protein.It is recommended to use one of the now common approaches to resurrect the last common ancestor of SpoIVA and Era (see e.g. 10.1146/annurev-biophys-070816-033631), and study its properties in an unbiased fashion (e.g., What sequence motifs emerge? What is its activity and substrate profile? Does it polymerize and/or form hydrolysis-dependent oligomers?).

We acknowledge that our use of the term “resurrection” could have implied a particular computational approach that has certainly been successful in objectively reconstructing ancestral proteins. It is clear that we have not used that approach; therefore, in the revised manuscript, we have completely eschewed the use of the term and instead refer to our mutagenesis study as a “reformulation” or “reengineering” of the enzyme’s active site.

The resurrection methods to which the reviewer refers can be successfully used for relatively closely related proteins to reconstruct a common ancestor: for example, we could have used it to reconstruct the ancestor of *B. subtilis* Era and *E. coli* Era and potentially “resurrect” it.

Instead, we needed to use our current approach in place of the methodology to which the reviewer refers because of the specific aspects of the evolutionary divergence between SpoIVA and Era:

1) The approach mentioned by the reviewer is well-suited for proteins that diverge from a distinct common ancestor. However, with SpoIVA and Era, there is an unusual situation in which there is no SpoIVA homolog that exists outside of sporulating Firmicutes. Its closest homolog is Era, which is a universally conserved bacterial protein and is an extant enzyme. Thus, the phyletic patterns of SpoIVA and Era indicate that SpoIVA was derived from an ancestral Era rather than the two being sister groups descending from a distinct reconstructable common ancestor. As we reported previously (Castaing et al., 2013) the simplest explanation is that SpoIVA evolved from Era, likely via gene duplication and followed by rapid divergence after paralog formation. The ancestor of SpoIVA, effectively, is Era itself. We realize that this evolutionary situation was not clearly explained in the text, since we simply referenced our earlier work. We have therefore included additional text that explains the evolution of SpoIVA (please see subsection “Amino acid substitutions in the nucleotide binding pocket could be responsible for the evolution of ATP binding specificity in SpoIVA”).

2) Further, SpoIVA has even acquired two C-terminal domain fusions that are not present in Era (now more prominently denoted in the schematic in Figure 1C). Moreover, in the process of diverging from Era SpoIVA has accumulated myriad substitutions along with unique insertions, which add considerable uncertainty to any complete reconstructions of the ancestral intermediates.

When all these are considered, it would result in too vast of a parameter space to analyze using the approach that the reviewer suggested and accordingly the number of reengineering steps involved would be currently infeasible. Therefore, despite the merits of the suggested approach, the example of SpoIVA and Era precludes employing it. Hence, we had to restrict ourselves to manipulating a limited but functionally clearly defined set of features of the TRAFAC GTPase domain in SpoIVA for which we have extensive experimental and computational evidence for how they function. Accordingly, we manipulated highly conserved residues known to participate in catalysis and, guided by sequence conservation and structure. In the process we accounted even for peripheral residues that we predicted to affect active site properties.

2) The authors use Era, "ancestral Era GTPase", and the ancestor interchangeably (e.g., subsection “The altered NKxD motif in G4 and SxE sequence in G5 mediate nucleotide specificity of SpoIVA”, but common throughout the manuscript). However, Era appears to be an extant enzyme that likely evolved independently after the emergence of SpoIVA, which may have further contributed to the specialization of the two proteins compared to a common ancestor. Careful separation between extant Era and SpoIVA and predicted common ancestors would avoid the potential for confusion.

We agree that the different labels for the same protein can be confusing, and therefore have changed most references to the protein as simply “Era”. As mentioned above, Era is indeed an extant enzyme, just like SpoIVA, but it certainly did not evolve after SpoIVA emerged. Furthermore, Era and SpoIVA do not appear to share a distinct common ancestor in the typical sense. Rather, Era is a universally conserved bacterial protein, while SpoIVA is entirely restricted to the clade of sporulating Firmicutes. Thus, the simplest model is that Era is indeed the ancestor and that SpoIVA emerged, likely via gene duplication, from Era and subsequently diverged away from Era. As mentioned above, we have now included a much more detailed description in the text of our previous work (subsection “Amino acid substitutions in the nucleotide binding pocket could be responsible for the evolution of ATP binding specificity in SpoIVA”) that explains this evolutionary relationship between SpoIVA and Era.

3) If the targeted changes in the primary active site motifs fully recapitulate the evolutionary and functional differences between SpoIVA and Era, one may predict that the corresponding changes in Era would result in SpoIVA-like properties. This would be a crucial experiment to consider for this study.

This is a bold proposal! We had initially considered performing this test, but ultimately concluded that it would be outside the scope of this study because although we initially argued that these residues were necessary for preferential ATPase activity, we were not prepared to state that they were sufficient to distinguish between ATP and GTP. That said, we decided to try the reviewer’s suggested experiment, and the results are now displayed in a new Figure 2A. Changing the “NKxD” motif of Era to “NSxR” found in SpoIVA, and introducing a Glu to Era to introduce an “SxE” motif similar to that of SpoIVA remarkably (we think) resulted in preferential hydrolysis of ATP compared to GTP when Era was stimulated. We thank the reviewer for challenging us to perform this experiment!

4) While the type of mutations introduced into SpoIVA was guided by sequence comparison with Era-type GTPases, it appears that the authors chose the mutant they refer to as the resurrected ancestor based on the activity profile that matches most closely the catalytic characteristics of Era, and not necessarily based on the phylogenic analysis. This practice introduces bias that may hamper the analysis concerning the evolution of substrate specificity.

As explained above, phylogenetic analysis similar to what the reviewer proposed would be feasible when the proteins are relatively closely related, but not at the evolutionary divergence between SpoIVA and Era. SpoIVA harbors, in addition to amino acid changes, two C-terminal domains that are not present in Era. To get around this insurmountable issue, we made various intermediate states and chose the activity state that most closely resembled Era, the protein from which SpoIVA likely evolved.

In this context, it is also not clear why the authors assume that the last common ancestor of SpoIVA and Era had Era-like hydrolysis efficiencies and substrate preference.

As explained above, Era itself is the most parsimonious likely ancestor to SpoIVA. SpoIVA is undetectable outside of the Firmicutes phylum and SpoIVA is most closely related to Era. Thus, we are faced with the unusual situation where Era itself gave rise to the extant SpoIVA- not that SpoIVA and Era had a distinct last common ancestor from which each diverged as descendent sister groups. Since almost all members of the TRAFAC family of GTPases (and certainly proteins in the Era lineage of enzymes other than SpoIVA) possess GTPase activity, we humbly suggest that it is not unreasonable to presume that the ancestor of SpoIVA would possess a substrate preference for GTP, not ATP, and that its enzymatic activity would closely reflect that of an Era GTPase.

5) The authors also did not consider the potential impact the register shift in G5 between SpoIVA (sequence SxE) and Era (sequence SA) may have, only focusing on the sequence register found in SpoIVA. It is conceivable that the additional residue in this motif arose to accommodate an ATP preference and ATPase-driven mechanism.

There may be a misunderstanding regarding our strategy for employing the alanine substitution and where exactly the glutamate resides in SpoIVA. To clarify, across the TRAFAC GTPases, the sequence simply happens to be “SA”, wherein the Ala does not contribute to catalysis or recognition of the purine in any way. Thus, the “SxE” equivalent sequence in Era is “SAx”, where the second “x” just happens to be an “A” in most TRAFAC GTPases. In Era it tends to be most frequently C or A which can provide the same small hydrophobic sidechain to effectively play a role no different from the A in the other GTPases. In SpoIVA, the E in the third position is an evolutionarily constrained residue (preferentially polar than hydrophobic). In certain groups of TRAFAC GTPases, the position after SA can be conserved (for example, Ras has a K) but in Era, the position following “SA” is not under any evolutionary constraint. The preference for a polar residue at that position appears to have therefore emerged specifically in the SpoIVA lineage. From a structural viewpoint, examination of TRAFAC GTPases indicates that, especially in the dimeric state, the third position can be proximal to the purine of the bound nucleotide thereby potentially affecting nucleotide specificity of the enzyme. To inactivate the E in SpoIVA, we employed a strategy common in site-directed mutagenesis, which is to simply substitute a residue of interest with A (which harbors a short sidechain and is unlikely to cause gross structural changes) to produce “SxA”. This strategy was NOT an effort to move the A one residue over, as the reviewer seems to have interpreted. Rather, it was to substitute the polar residue “E” in the SxA sequence. We hope this clarifies our approach and logic.

In the revised manuscript, to evolve ATPase activity in Era as suggested by the reviewer, we introduced a glutamate after the conserved “A” in Era (to produce “SAE”, thereby recapitulating the “SxE” found in SpoIVA) because (1) we predicted that it resides in the binding site of SpoIVA, and (2) we predict that it is a new state that exists on top of ground GTPase state. In other words, we propose that ATP binding is an acquired state. Thus, we did not introduce a register shift, and the alanine substitution strategy was completely unrelated to the coincidental “A” that occupied the “x” position in Era. To avoid confusion, we have now referred to the corresponding Era sequence as “SAx” to indicate that we are not introducing a register shift and have added sentences that more clearly compare the sequences in this region and explains our mutagenesis strategy (Results).

6) It is not clear whether the defects in polymerization characteristics of the SpoIVA-resurrected mutant can be attributed to the lower ATPase activity compared to wild-type SpoIVA (or other subtle changes introduced by the mutations). It is also not clear whether SpoIVA-resurrected forms a similar ATPase-dependent intermediate that was oberserved with wild-type SpoIVA (at protein concentrations below the threshold for polymerization). This is an important experiment since the differences between ATP and GTP in their capacity to drive intermediate formation could be explained by the different ATPase vs GTPase efficiencies of the wild-type protein; SpoIVA-resurrected has overall lower catalytic activity compared to wild-type, and it does not support polymerization. Hence, there is a possibility that the observed differences can be attributed to catalytic efficiency (maybe in addition to the nature of the nucleotide base).

We agree that this is an important experiment. We have now tested if the SpoIVA^GTPase^ variant forms the assembly intermediate at a low protein concentration (under non-polymerization conditions). The results are presented in a new Figure 5B-C and indicate that (1) this variant is capable of forming the ADP-associated assembly intermediate, albeit at an expected lower efficiency than WT, and (2) that the variant does NOT form the intermediate in the presence of GTP. Taken together, the results allow us to conclude that the SpoIVA^GTPase^ variant is able to utilize ATP to polymerize but unable to utilize GTP to polymerize because it fails to form the critical NDP-bound intermediate that is required for the assembly of the nucleotide-free polymer. We thank the referee for suggesting this experiment.